



# Upper tropospheric slightly ice-subsaturated regions: Frequency of occurrence and statistical evidence for the appearance of contrail cirrus

Yun Li[1,2], Christoph Mahnke[1], Susanne Rohs[1], Ulrich Bundke[1], Nicole Spelten[2], Georgios Dekoutsidis[3],
Silke Groß[3], Christiane Voigt[3,4], Ulrich Schumann[3], Andreas Petzold[1] and Martina Krämer[2,4]

[1]Forschungszentrum Jülich, Institute of Energy and Climate Research – Troposphere (IEK-8), Jülich, Germany
[2]Forschungszentrum Jülich, Institute of Energy and Climate Research – Stratosphere (IEK-7), Jülich, Germany
[3]Deutsches Zentrum für Luft- und Raumfahrt (DLR), Institut für Physik der Atmosphäre, Oberpfaffenhofen, Germany
[4]Johannes Gutenberg-Universität, Institute of Atmospheric Physics, Mainz, Germany

*Correspondence to*: Yun Li (yun.li@fz-juelich.de) and Martina Krämer (m.kraemer@fz-juelich.de)

**Abstract.** Microphysical, optical, and environmental properties of contrail cirrus and natural cirrus were investigated by applying a new, statistically based contrail–cirrus separation method to 14.7 hours of cirrus cloud measurements during the airborne campaign ML-CIRRUS in Central Europe and the Northeast Atlantic flight corridor in Spring 2014. We find that pure contrail cirrus appears frequently at the aircraft cruising altitude (CA) range with ambient pressure varying from 200 to 245 hPa. They exhibit a higher median ice particle number concentration ($N_{ice}$), a smaller median mass mean radius ($R_{ice}$), and lower median ice water content (IWC) (median: $N_{ice}$ = 0.045 cm$^{-3}$, $R_{ice}$ = 16.6 µm, IWC = 3.5 ppmv), and they are optically thinner (median extinction coefficient Ext = ~ 0.056 km$^{-1}$) than the cirrus mixture of contrail cirrus, natural in situ-origin and liquid-origin cirrus found around the CA range (median: $N_{ice}$ = 0.038 cm$^{-3}$, $R_{ice}$ = 24.1 µm, IWC = 8.3 ppmv, Ext = ~ 0.096 km$^{-1}$). The lowest and thickest cirrus, consisting of a few large ice particles, are identified as pure natural liquid-origin cirrus (median: $N_{ice}$ = 0.018 cm$^{-3}$, $R_{ice}$ = 42.4 µm, IWC = 21.7 ppmv, Ext = ~ 0.137 km$^{-1}$). Furthermore, we observe that, in particular, contrail cirrus occurs more often in slightly ice-subsaturated instead of merely ice saturated to supersaturated air as often assumed, thus indicating the possibility of enlarged contrail cirrus existence regions. The enlargement is estimated, based on IAGOS long-term observations of relative humidity with respect to ice ($RH_{ice}$) aboard passenger aircraft, to be approximately 10% for Europe and the North Atlantic region with the $RH_{ice}$ threshold for contrail cirrus existence decreased from 100% to 90% $RH_{ice}$ and a 4-hour lifetime of contrail cirrus in slight ice-subsaturation assumed. This increase may not only lead to a non-negligible change in contrail cirrus coverage and radiative forcing but also affect the mitigation strategies of reducing contrails by rerouting flights.

## 1 Introduction

The global aviation sector makes up approximately 5% of anthropogenic global warming (Grewe et al., 2021; Klöwer et al., 2021). Contrail cirrus is one of the largest radiative forcing components of aviation (Lee et al., 2009; 2021) with uncertainties



arising from many sources, including limited knowledge of cirrus cloud properties, spatial coverage and life cycle (Schumann and Heymsfield, 2017; Kärcher, 2018; Burkhardt et al., 2018). Contrail cirrus comprises line-shaped contrails in the wake of high-flying aircraft and thin cirrus patches resulting from the dispersion of long-living contrails. Only a few models account for the water emitted from the aircraft causing contrails in slightly subsaturated air and for the ice water content in contrails

during their life cycle extending their persistence (Schumann, 2012). Instead, models often estimate contrail cirrus coverage based on simplified contrail ageing and spreading mechanisms in ice-supersaturated regions (ISSRs) (Burkhardt et al., 2010; Burkhardt and Kärcher, 2011).

Contrails form when hot and humid aircraft exhaust mixes rapidly with cold and humid ambient air so that the humidity in the exhaust gases exceeds liquid water solution (Appleman, 1953; Schmidt, 1941; Schumann, 1996). In such air supersaturated

with respect to liquid water ($RH_w > 100\%$), aerosol particles emitted from the aircraft (combustion soot and sulphuric acid-water droplets) or pre-existing in the in-mixed ambient air become activated to form water droplets that freeze subsequently to contrail ice particles. According to the purely thermodynamic Schmidt-Appleman criterion (SAC), the threshold temperature for contrail formation depends on ambient air pressure and humidity, on the amount of water and heat emitted by the aircraft per fuel mass and on the aircraft engine's overall propulsion (Schumann, 1996; Jensen et al., 1998). After reaching the ambient

temperature by mixing with the surrounding air, the contrails grow or shrink in size depending on ambient humidity. If the ambient relative humidity remains supersaturated with respect to ice ($RH_{ice} > 100\%$), contrails grow in ice water content and can persist for up to 5 h or even longer (Gierens and Vázquez-Navarro, 2018; Schumann and Heymsfield, 2017), and may spread and evolve into thin cirrus layers. Otherwise, contrail ice particles sublimate and dissipate on a timescale dependent on their size and the ambient air $RH_{ice}$ (Schumann, 2012).

A robust estimation of contrail cirrus' radiative effect depends largely on their optical properties (related to their microphysical properties and age) and geographical appearance. Young contrails can exert an instantaneous radiative forcing to warm and cool the atmosphere that is three orders of magnitude larger than their net warming effect (Gierens et al., 2020). The microphysical features of contrail cirrus at different plume ages observed from various airborne campaigns were compiled and described in Schröder et al. (2000), Schumann et al. (2017) and Chauvigné et al. (2018). Fresh contrails are characterised by

an ice crystal number concentration ($N_{ice}$) of thousands of ice crystals per cubic centimetre in size up to a few micrometres in diameter, as observed in an approximately 2-minute-old plume (Petzold et al., 1997). Two to five minutes old contrails were frequently measured (*e.g.* Voigt et al., 2011; Gayet et al., 2012). Here, contrail ice crystal number concentrations were diluted to 100 to 400 per cubic centimetre and ice crystal diameters increased to 4 to 10 µm due to condensational growth (Jeßberger et al., 2013; Bräuer et al., 2021). Slightly older contrails at a maximum plume age of 30 minutes dilute further by the inmixing

of ambient air down to less than hundreds of ice crystals that have grown to tens of micrometres (Schröder et al., 2000). The peculiar high $N_{ice}$ of small ice particles makes young contrails easy to be distinguished from natural cirrus. At an even later stage, $N_{ice}$ of contrail cirrus further decreases significantly to a few ice particles per cubic centimetre or less, with particle sizes being 2–3 orders of magnitude larger, becoming similar to natural cirrus and making the discrimination between contrail and natural cirrus difficult. Contrail cirrus are generally characterized by low ice water content (IWC) ranging from 0.1 to about





10 mg/m³ (Schumann et al., 2017), like natural cirrus of in situ-origin whose ice crystals have formed and grown in an ice cloud only environment (Luebke et al., 2016; Krämer et al., 2020). Different from contrail cirrus and in situ-origin cirrus, liquid-origin cirrus often yield higher IWC (Krämer et al., 2016; Krämer et al., 2020) because their ice crystals originally form as liquid drops in a warmer atmosphere ($T_{amb} > 235$ K), which subsequently freeze while being lifted into the cirrus temperature region of the atmosphere.

The fact that contrails often coexist with natural cirrus and become embedded within thin or subvisible cirrus (Kübbeler et al., 2011; Gierens, 2012; Unterstrasser, 2017) makes it challenging to discriminate between aged contrails and natural cirrus, thus impeding clarifying contrail cirrus' contribution to the radiative balance. Chauvigné et al. (2018) has employed a principal component analysis method to distinguish between contrail cirrus particles at different ages and natural cirrus measured during the CONCERT 2018 campaign (Voigt et al., 2010), which is successful because contrails sampled during the CONCERT

campaign were rather young and more recognisable compared to natural cirrus. However, not all required optical and microphysical parameters can be obtained from single aircraft campaigns to apply this method widely, and the CONCERT dataset is small, around 4.0 h of contrail and natural cirrus sampling time in total (Kübbeler et al., 2011).

A common assumption on the conditions for contrail cirrus formation and evolution is that contrail cirrus occurs and persists merely in ISSRs (Kärcher, 2018). In fact, contrails and contrail cirrus were also observed in ice-subsaturated air not only during

contrail-dedicated research flights (Kübbeler et al., 2011; Voigt et al., 2011; Gayet et al., 2012; Schumann et al., 2017; Chauvigné et al., 2018) but also from IAGOS commercial aircraft observations in the North Atlantic region (Petzold et al., 2017). Apart from a high number of small contrail ice particles, large particles (ice particle diameter $D_p > 100$ µm) were also detected, but at relatively low concentrations (Voigt et al., 2010; Kübbeler et al., 2011; Chauvigné et al., 2018). Such large ice crystals were also observed in contrail cirrus during the ML-CIRRUS campaign (Voigt et al., 2017). However, attention to

contrail cirrus in ice-subsaturated environments and the role that large ice particles play in contrail cirrus is raised only by Kübbeler et al. (2011) and Schumann (2012). Kübbeler et al. (2011) discussed that the subsaturation feature observed in contrail cirrus during the CONCERT campaign is accompanied by the sublimation of those large ice particles, which might be sedimented from higher levels after being formed under ISSRs. As the contrail cirrus dataset is limited to only a few segments of several flights, it could not be corroborated that the existence of contrail cirrus in ice-subsaturated environments is a common

feature. But Schumann and Graf (2013) found it necessary to reduce the critical humidity above which contrails form to a value below ice saturation to model contrail occurrence and their longwave radiative forcing in agreement with multi-year satellite observations over the North and South Atlantic.

Currently, the prevalent strategy for contrail avoidance is to reroute the aircraft around ice-supersaturated regions by flying at slightly higher or lower altitudes to avoid contrail formation or minimise contrail radiative forcing (Teoh et al., 2020a; Niklaß

et al., 2021). Teoh et al. (2020a) showed that focusing on the avoidance of strong contrails, the so-called big hits, reduces the radiative forcing effect because only for these cases, the "saved" radiative forcing overrules the additional $CO_2$ emitted during the rerouting of the aircraft. Gierens et al. (2020) showed that the formation of contrails can be predicted with some success, but there are problems in predicting contrail persistence due to limited knowledge about the occurrence of air masses around



ice saturation. Particularly from this study, it becomes evident that further knowledge about the distribution of air masses
around ice saturation and the resulting properties of aircraft-induced cirrus and natural cirrus is required (Teoh et al., 2022).

In this study, we investigate a larger dataset of 14.7 h cirrus cloud sampling obtained during the ML-CIRRUS 2014 campaign (Sect. 2.1) than the 4-hour CONCERT dataset. With commonly available parameters describing the microphysical properties of cirrus, such as $N_{ice}$, ice crystal sizes and IWC, we adopt a simpler statistical approach to separate aviation-induced cirrus from natural cirrus compared with Chauvigné et al. (2018). It consists of the SAC, the most frequent aircraft cruising altitude
range and a newly developed aircraft exhaust plume detection algorithm (Mahnke et al., 2022) to differentiate aged contrail cirrus (> 0.5 h lifetime, Schumann et al., 2017; Voigt et al., 2017) and natural cirrus (Sect. 2.3). We show step by step the sharpened differentiation of contrail cirrus from natural cirrus and report on their microphysical properties and occurrence conditions (Sect. 3.1–3.2). In addition, we analyse the humidity of the environments of contrail and natural cirrus (Sect. 3.3). Based on these observations, we simulate the lifetime of ice particles that have similar microphysical properties to the contrail
cirrus sampled during ML-CIRRUS in slightly ice-subsaturated environments. Furthermore, we inspect 15 years of $RH_{ice}$ measurements aboard passenger aircraft in the IAGOS global monitoring framework to shed light on how the existence of contrail cirrus in environments with $RH_{ice} \geq 90\%$ might influence contrail mitigation (Sect. 4).

## 2 Datasets and methods

### 2.1 ML-CIRRUS dataset

The Mid-Latitude CIRRUS (ML-CIRRUS) campaign was conducted from Oberpfaffenhofen, Germany to probe cirrus clouds over Central Europe and the Northeast Atlantic region in March and April 2014 (Voigt et al., 2017). The High Altitude and Long-Range Research Aircraft (HALO, Krautstrunk and Giez, 2012) was deployed to investigate the formation mechanism, life cycle and climate impact of natural cirrus and aircraft-induced cloudiness. Excluding test flights and the ones with strong instrumental issues, 12 of 17 research flights with the focus on natural and contrail cirrus, as listed in Table S1, are considered
here for studying the microphysical properties of contrail cirrus and mid-latitude natural cirrus, which serve as the basis to distinguish contrail cirrus from natural cirrus.

The ML-CIRRUS dataset (Voigt et al., 2017) includes the parameters important for cloud characterisation — in situ relative humidity with respect to ice $RH_{ice}$, ice water content IWC, ice particle number concentration $N_{ice}$ and mass mean radius $R_{ice}$. The in situ $RH_{ice}$ was calculated using water vapour mixing ratios measured using the tuneable diode laser hygrometer SHARC
(Meyer et al., 2015), ambient temperature $T_{amb}$ and pressure measurements provided by the Basis Halo Measurement and Sensor System (BAHAMAS) (Mallaun et al., 2015; Giez et al., 2017). The in situ $RH_{ice}$, water vapour and temperature measurements were compared with other instruments on board and model data from the European Centre of Medium-range Weather Forecasting (ECMWF) by Kaufmann et al. (2018). No systematic instrument bias in either water vapour or temperature was identified in the upper troposphere.



Cloud measurements were performed with using cloud spectrometer NIXE-CAPS (New Ice eXpEriment: Cloud and Aerosol Particle Spectrometer, later referred as NIXE) at a time resolution of 1 Hz with the instrument mounted under the aircraft wing (Krämer et al., 2016; Luebke et al., 2016). As a combination of the two instruments NIXE-CAS-DPOL (Cloud and Aerosol Spectrometer with Detection of POLarization) and NIXE-CIPg (Cloud Imaging Probe — grayscale), NIXE measures $N_{ice}$ in the particle diameter ($D_p$) range of 0.61–937 µm. Only particles $D_p > 3$ µm are considered for cloud measurements, while

smaller particles are classified as aerosols. In fresh contrails, particle sizes can be smaller than 3 µm, but for consistency and comparability, the lower threshold $D_p = 3$ µm is maintained in the analysis of contrails and contrail cirrus. NIXE gives a total uncertainty of ± 20% (Meyer, 2012) in particle number concentration measurement. IWC is derived from the ice particle size distribution ($PSD_{ice}$) in the $D_p$ range of 3–930 µm. How the IWC is determined from a mass-dimension relation and the robustness of the IWC have been stated in Krämer et al. (2016), Luebke et al. (2016) and Afchine et al. (2018). The lower IWC

detection limit of NIXE is 0.15 ppmv (parts per million by volume). The ice crystal mass mean radius $R_{ice}$ in µm is calculated

with $R_{ice} = 1.e^4 \times \left(\frac{1.e^{-6}IWC}{N_{ice}} \times \frac{3}{4\pi\rho}\right)^{1/3}$, where $IWC$ is in mg/m$^3$, $\rho$ is 0.92 g cm$^{-3}$ and $N_{ice}$ is the total number concentration

of ice crystals ($D_p > 3$ µm) in cm$^{-3}$.

Additional parameters for discriminating contrail and natural cirrus are total aerosol particle number concentration, total reactive nitrogen $NO_y$ mixing ratio and airborne lidar $RH_{ice}$. Here, the measurements are summarised below (see Voigt et al.,

2017 for details).

The total aerosol particle number concentration was measured by the instrument AMETYST (Voigt et al., 2017), which is a combination of four condensation particle counters (CPCs) measuring total and non-volatile aerosols in the size range of 4 nm–2 µm. The uncertainty of the CPCs of AMETYST is in the typical CPC uncertainty range, which is estimated to be of the order of 10% (Petzold et al., 2011; 2013). $NO_y$ was measured by the instrument AENEAS (Ziereis et al., 2000) by catalytically

converting $NO_y$ to nitrogen monoxide NO on a gold surface heated at 300 °C. The converted NO will then be directly detected with chemiluminescence technique. AENEAS has an $NO_y$ detection range of 5 parts per trillion by volume (pptv) to 60 parts per billion by volume (ppbv) (Voigt et al., 2017), with an overall uncertainty of 30% or 40 pptv.

The airborne lidar $RH_{ice}$ is derived from water vapour measurement in the 935 nm absorption band of $H_2O$ by the lidar WALES and ambient temperature from ECMWF (Wirth et al., 2009; Groß et al., 2014). For retrieving cirrus clouds from the remote-

sensing technique, only the particles producing a back-scattering ratio (BSR) greater than 3 and having a depolarisation ratio greater than 20% at $T_{amb} < 235$ K are interpreted as cirrus cloud particles (Urbanek et al., 2018). Note that SHARC measures water vapour concentrations at aircraft positions, WALES obtains atmospheric cloud columns with its laser penetrating through clouds from the cloud top or bottom. The two instruments do not measure water vapour in parallel. Therefore, the in situ $RH_{ice}$ and lidar $RH_{ice}$ are not from the same clouds. However, the intercomparison of in situ and lidar $RH_{ice}$ measurements inside

cirrus clouds promotes the evaluation of the robustness of the in situ $RH_{ice}$ dataset and uncertainties related to the quality of the $T_{amb}$ dataset.



## 2.2 RH$_{ice}$ dataset from IAGOS passenger aircraft

The RH$_{ice}$ dataset spanning from 1995 to 2010, based on the Measurement of Ozone and Water Vapour on Airbus In-service Aircraft (MOZAIC) programme, is used for the analysis of the RH$_{ice}$ distribution in air masses in the northern mid-latitudes.

The MOZAIC programme (Marenco et al., 1998) was initiated in August 1994 and was carried on within the new European Research Infrastructure IAGOS (In-service Aircraft for a Global Observing System, https://www.iagos.org/) in 2011 (Petzold et al., 2015). The measurement of atmospheric trace gases and aerosol particles is conducted by autonomous instruments installed on commercial passenger aircraft. Up to now, over 63,000 flights contribute to a global-scale dataset of water vapour and RH$_{ice}$ in the upper troposphere and lower stratosphere (Petzold et al., 2017; Petzold et al., 2020; Reutter et al., 2020).

The dataset used for this study has aerial boundaries of 40°–60° N covering the North Atlantic (65°–5°W) and Europe (5°W–30°E). It contains temperature, pressure and RH$_{ice}$ measurements. RH$_{ice}$ is directly measured by the IAGOS Capacitive Hygrometer (ICH), which is calibrated before and after being deployed onto the aircraft. The details regarding the principles of the ICH sensor and sensor calibration as well as the procedures to determine the ambient air temperature from the sensor temperature can be found elsewhere (Neis et al., 2015; Smit et al., 2014). The data were quality checked and compared to

ERA-Interim reanalysis data (Dee et al., 2011) to be reliable for scientific studies (Neis et al., 2015; Petzold et al., 2020; Reutter et al., 2020). A more detailed application of the dataset was described in the study of ice-supersaturated air masses in the northern mid-latitudes (Petzold et al., 2020). Here, the occurrence fractions of air masses at different RH$_{ice}$ thresholds are determined from this dataset and utilized for the discussion about the potential influence on contrail avoidance (Sect. 3.5).

## 2.3 Contrails and contrail cirrus detection

Data suitable for statistically analysing the microphysical properties of cirrus induced by aircraft emissions or by atmospheric dynamic systems should meet the following criteria:

1. ambient pressure p < 350 hPa, which constrains the pressure altitude to be higher than ~ 8.1 km under standard atmospheric conditions. This is also the common cruising altitude of commercial airplanes.
2. ambient temperature T$_{amb}$ < 235 K, which is the cirrus formation temperature region.

### 2.3.1 The Schmidt-Appleman criterion (SAC)


To determine the potential for contrail formation in the air masses meeting the above thresholds, air mass thermodynamic properties are analysed by applying the Schmidt-Appleman criterion (SAC). The SAC at aircraft pressure level *p* depends on the gradient *G* of the mixing line (see Fig. 1) (Schumann, 1996):

$$G = \frac{EI_{H_2o}\, c_p p}{\varepsilon Q(1-\eta)}$$









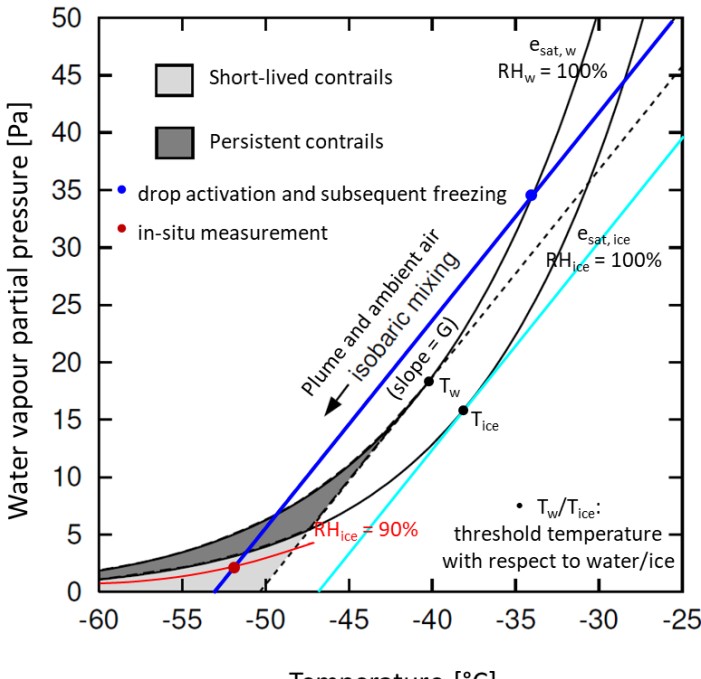

**Figure 1.** Water vapour saturation partial pressure with respect to liquid water ($e_{sat, w}$) and ice ($e_{sat, ice}$) as a function of temperature. The blue line represents the isobaric mixing of the aircraft exhaust plume with the surrounding ambient air at a gradient of G along the black arrow. During the isobaric mixing, liquid drops form when surpassing water saturation; the drops freeze subsequently at ~ 235 K. The red dot represents an in situ measured cloud sample that marks the ending point of the cloud sample's mixing. $T_w$ is the threshold temperature for contrails to form. This occurs when the isobaric mixing line (the dashed line) just touches the water saturation curve. In this case, contrails persist in ice-supersaturated environments and live only shortly in ice-subsaturation, as indicated by the dark and light grey areas, respectively. The cyan line shows a situation where the aircraft exhaust air parcel just reaches ice saturation so that contrail ice particles might form directly from the gas phase heterogeneously or homogeneously at the corresponding ice supersaturation. The red curve at $RH_{ice}$ = 90% represents the proposed lower $RH_{ice}$ threshold of persistent contrails in this work (Section 3.3).

where $p$ is the ambient air pressure, $c_p$ is the isobaric heat capacity of air (1004 J kg$^{-1}$ K$^{-1}$), $\varepsilon$ is the ratio of molar masses of

water and dry air (0.622). A water vapour emission index $EI_{H_2o}$ = 1.25 kg/kg-fuel, fuel heat capacity $Q$ = 43.2 MJ/kg-fuel for

convectional jet fuel (kerosene) and the overall propulsion efficiency $\eta$ = 0.31 are applied to the calculation. G (units Pa K$^{-1}$)

represents the gradient of the trajectory of aircraft exhaust air isobaric mixing with the surrounding ambient air — the blue

line in Fig. 1, where the dependence of the water vapour partial pressure on temperature in the isobaric mixture of an aircraft

plume and ambient air is illustrated (adapted from Fig. 3 in Schumann (1996)). The sampled air masses are assumed to be

released from aircraft engines and have undergone the isobaric mixing process while detraining into ambient air. $T_{amb}$ and

$RH_{ice}$ at an assumed measuring position (the red dot in Fig. 1), therefore, mark the ending point of an individual air parcel's

mixing line. If the mixing line touched or crossed the ice/liquid saturation curve, *e.g.,* the blue, dashed black or cyan lines in

Fig. 1, the measured cloud particle could have been very probably involved in the formation of contrails during its evolution.

Cirrus cloud particles sampled at thermodynamic positions not fulfilling SAC are considered irrelevant to contrail formation

and are treated as natural cirrus.





### 2.3.2 The most frequent aircraft cruising altitude (CA) range

To better discriminate between contrail and natural cirrus, we define another criterion by dividing the altitude range of the dataset fulfilling SAC into the most frequent aircraft cruising altitude (CA) range and the altitudes beyond. To determine the

CA pressure range, we surveyed the 15 years of IAGOS-MOZAIC pressure measurements over the North Atlantic and Europe regions aboard passenger aircraft. The occurrence probabilities of flight levels per 10-hPa bin are shown in Fig. 2. The most frequently visited atmospheric pressure levels are from 200 to 270 K. However, ice cloud properties at lower altitudes with pressures greater than 245 hPa show distinct behaviour from those at higher altitudes below 245 hPa and seem closer to those observed at positions not fulfilling SAC, implying that they could be natural cirrus or extensively aged contrail cirrus. Hence,

the pressure altitude range of 200–245 hPa is adopted as the CA range for the further analysis.

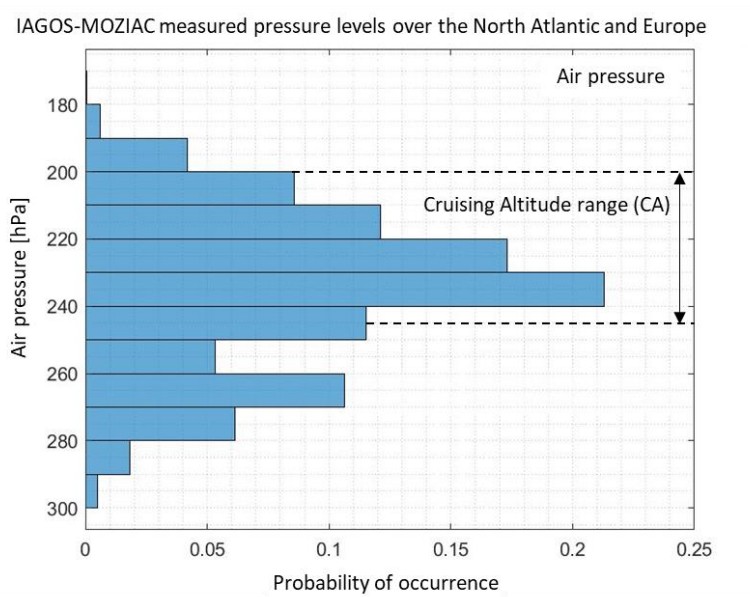

**Figure 2.** Occurrence fractions of passenger aircraft flight levels in pressure (unit: hPa) measured during the IAGOS-MOZAIC period (1996–2010) over the North Atlantic and Europe. The pressure range 200–245 hPa between the dashed lines is adopted as the most frequent aircraft cruising altitude (CA) range (see text for the determination of the boundary pressure thresholds).

### 2.3.3 Aircraft plume detection

The principle of the aircraft exhaust plume detection assumes that simultaneous enhancement of multiple products emitted

from aviation fuel combustion, here total aerosol particles and reactive nitrogen $NO_y$ measured by the instruments AMETYST and AENEAS (Sect. 2.1), serves as a clear marker for air masses influenced by aircraft exhaust (Schumann et al., 2002). In this work, the total aerosol particle number concentration and the mixing ratio of $NO_y$, which can be considered a passive tracer in aircraft plumes up to a plume age of 18 h, are used to detect aircraft exhaust plumes along the flights from the concurrent increased particle number concentrations and $NO_y$ mixing ratios in comparison to atmospheric background values

(Mahnke et al., 2022). Only the air masses containing ice particles detected at $T_{amb} < 235$ K are considered as contrail cirrus.





The current algorithm does not provide information about the age of the encountered plumes, but it allows us to identify whether a sampled air mass originated from aircraft exhaust.

## 3 Properties of contrail and natural cirrus

### 3.1 Cirrus cloud observations

The full ensemble of cirrus cloud properties ($N_{ice}$ and $R_{ice}$) observed during ML-CIRRUS as a function of ambient temperature is shown in Fig. 3 together with in situ $RH_{ice}$. The number of flight hours spent in cirrus clouds is 14.7 h. Figure 3a and b show the temperature dependence of $N_{ice}$ ($D_p > 3$ µm) and $R_{ice}$, respectively, binned in 1-K intervals and color-coded by the occurrence frequency that is normalised to the total counts in each temperature interval. One pronounced signature of contrail cirrus is the high $N_{ice} > 0.1$ or even $> 1$ cm$^{-3}$ between 208–220 K, which could be linked to aviation-induced cirrus (Petzold et

al., 2017; Schumann et al., 2017; Krämer et al., 2020) as the median $N_{ice}$ of the large climatology of cirrus shown by Krämer et al. (2020) is 0.03 cm$^{-3}$. In the same temperature range, $R_{ice}$ exhibits the highest occurrence frequency at a small mass mean radius around 20 µm or even lower. The occurrence frequency of $N_{ice}$ in relation to $R_{ice}$ is displayed in Fig. 3c for the whole cirrus dataset with the isolines representing IWC in ppmv, which means that the same IWC could arise from many small ice particles (the upper-left segment of the IWC isolines) or a few large ice crystals (the lower-right segment).

Contrail cirrus often appear in the upper left side of the banana-shaped $N_{ice}$–$R_{ice}$ relation (Fig. 3c), typical of high $N_{ice}$, small $R_{ice}$ and low IWC, while natural cirrus more typically cluster in the middle and lower right parts with low $N_{ice}$, larger $R_{ice}$ and high IWC. The 50th (grey) and 90th (black) percentile contours indicate a pronounced occurrence of contrail cirrus, with IWC mostly below 10 ppmv. The 10 ppmv IWC isoline also roughly sets in situ-origin cirrus apart from liquid-origin cirrus with higher IWC. This classification of cirrus origins was applied to the ML-CIRRUS measurements by Luebke et al. (2016) and

is replotted in Fig. S1 (see Supplement), where it can be seen that in situ-origin cirrus appears more frequently with $R_{ice} < 30$ µm and IWC < 10 ppmv, while liquid-origin cirrus shows exactly the opposite.

The occurrence frequency of in-cloud $RH_{ice}$ in relation to temperature, as depicted in Fig. 3d, shows one maximum at around 90% $RH_{ice}$ in the temperature range of 208–220 K, which corresponds to the temperature range showing contrail cirrus signals in Fig. 3a and b. The observed occurrence of contrail cirrus in slight subsaturation with respect to ice observed in ML-CIRRUS

is consistent with what was reported in Kübbeler et al. (2011) based on the CONCERT dataset. A similar feature is reported from the first analysis of the IAGOS $RH_{ice}$ and cloud dataset in the North Atlantic Flight Corridor for the years 2014 and 2015 (Petzold et al., 2017).

In the following we will discriminate between contrail and natural cirrus and obtain an understanding of the occurrence of contrail cirrus regarding spatial probabilities, cloud properties as well as their favourable atmospheric conditions.




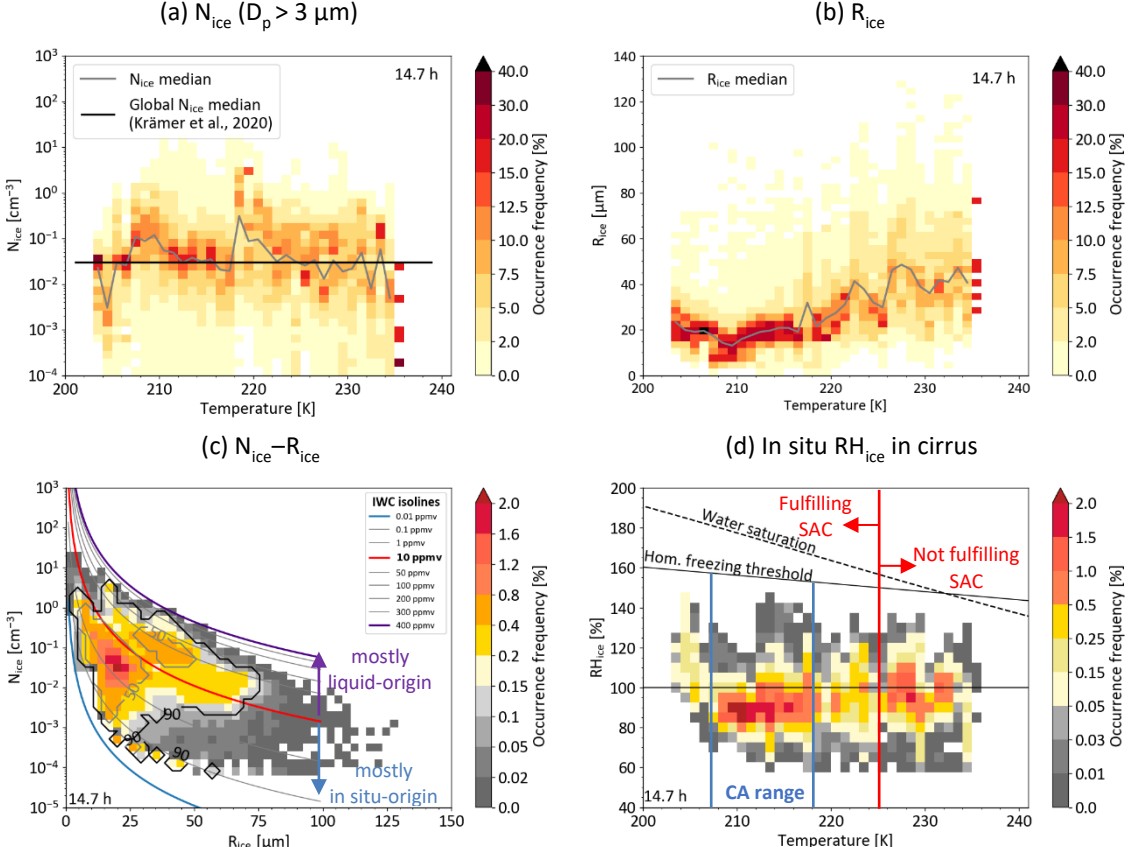

**Figure 3.** Overview of the cirrus cloud properties measured in Central Europe and the Northeast Atlantic flight corridor during the ML-CIRRUS research aircraft campaign in Spring 2014. (a): Occurence frequency of ice particle number concentrations ($N_{ice}$) for all cirrus crystals larger than 3 μm in diameter ($D_p$), binned in 1-K intervals. The grey line shows the median $N_{ice}$ in single temperature intervals. The horizontal bar indicates the $N_{ice}$ median from in situ $N_{ice}$ global climatology (Krämer et al., 2020). (b): The same as (a) but for the mass mean radii $R_{ice}$ of cirrus particles. (c): Normalised occurrence frequency of $N_{ice}$ as a function of $R_{ice}$. Coloured curves are ice water content IWC isolines in parts per million by volume (ppmv). The same amount of IWC could consist of many small ice particles pointing to the left end of the isoline or few large ice crystals to the right end. The grey and black contours enclose 50% and 90% of the most frequently occurring cloud particles. Ice crystals with IWC < 10 ppmv are mostly in situ-origin cirrus, while those with IWC > 10 ppmv are mostly liquid-origin cirrus. (d): Normalised occurrence frequency of in situ $RH_{ice}$ in cirrus clouds. The water saturation (Murphy and Koop, 2005) and homogeneous freezing threshold (Koop et al., 2000) are added. The red vertical line marks the temperature threshold for possible contrail formation, calculated from the Schmidt-Appleman criterion (SAC). The most frequent aircraft cruising altitude boundaries are marked by the blue vertical lines and correspond to a pressure range of 200–245 hPa, 207–218 K.

## 3.2 Cirrus classification

### 3.2.1 Cirrus differentiated by the Schmidt-Appleman criterion (SAC)

Our attempt at SAC calculation divides the ML-CIRRUS cirrus dataset discussed in the previous section into two categories: the dataset of cloud particles sampled under conditions fulfilling SAC (SAC⁺, ~ 11.2 hours) and the complementary dataset not fulfilling SAC (SAC⁻, ~ 3.5 hours); see Fig. 1d for details. As discussed in Sect. 2.3.1 and shown in Fig. 1, a cloud sample





fulfilling SAC is considered as being very possibly involved in the formation of contrails during its evolution, and the one
measured at thermodynamic positions failing SAC is considered irrelevant to contrail formation and are regarded as natural
cirrus.

Note, however, that natural cirrus could also be included in the SAC$^+$ group, meaning that SAC alone is not a sufficient criterion
to identify contrail cirrus, while cirrus detected in the SAC$^-$ group can be unambiguously attributed to natural cirrus. Despite
the limited differentiation between contrail and natural cirrus by using SAC alone, we will first discuss the differences in the
microphysical properties between the SAC$^+$ and SAC$^-$ datasets. The sharpening of the separation of the full ensemble into
aviation-influenced cirrus and natural cirrus by adding another criterion of the most frequent cruising altitude range will then
be discussed in Sect. 3.3.

### 3.2.1.1 Microphysical properties in the cirrus fulfilling SAC and in natural cirrus

Figure 4a and b display the $N_{ice}$–$R_{ice}$ relations for the SAC$^+$ and SAC$^-$ datasets, respectively, with occurrence frequency
normalized to the total number of measurements in each dataset. Small ice crystals in higher concentrations are mainly found
in the SAC$^+$ group. In the SAC$^+$ dataset, the median $N_{ice} = 0.04$ cm$^{-3}$ doubles the median value of the SAC$^-$ dataset. On the
contrary, the median $R_{ice}$ of the SAC$^+$ group (20.6 µm) is only half of the value (42.4 µm) of the SAC$^-$ counterpart. $N_{ice}$ in
the SAC$^+$ dataset reaches values as high as 20 cm$^{-3}$ for $R_{ice}$ below 30 µm, while $N_{ice}$ in the SAC$^-$ dataset is mostly below 1 cm$^{-3}$ with most particles larger than 30 µm. The highest occurrence frequencies in the SAC$^+$ and SAC$^-$ datasets (enclosed by the
50th percentile contours) dwell on the lower and upper sides of the 10 ppmv IWC isoline, respectively.

Besides the fact that in the SAC$^+$ dataset both natural and contrail cirrus can be found, the SAC$^-$ group contains only natural
cirrus. The two groups largely correspond to cirrus formed from different mechanisms, namely in situ- and liquid-origin cirrus.
Most in situ-origin (IWC < 10 ppmv) and some liquid-origin cirrus (IWC > 10 ppmv) are found in the SAC$^+$ group; conversely,
most of the liquid-origin cirrus and a small part of the in situ-origin cirrus are in the SAC$^-$ counterpart. Contrail cirrus belongs
to the in situ-origin cirrus type, since it appears in the temperature range $T_{amb} < 235$ K, either without pre-existing cirrus
(contrail cirrus) or superimposed on an existing cirrus (embedded contrails). As embedded contrails, they could also appear as
the liquid-origin cirrus.

The contrast between the SAC$^+$ and the SAC$^-$ groups is illustrated in Fig. 4c showing the differences between Fig. 4a and b.
The reddish area indicates that ice crystals measured in the environments satisfying SAC are prone to contrail cirrus, whereas
the bluish area, in contrast, shows ice crystals that are linked to natural cirrus.


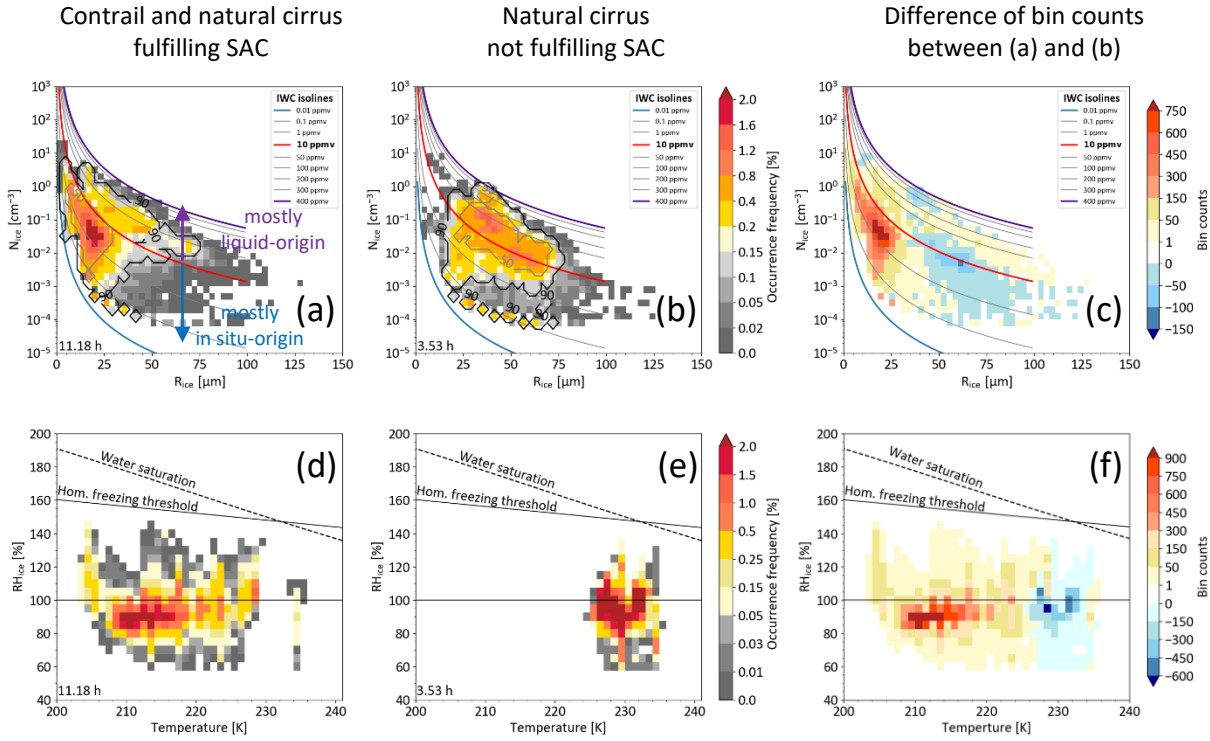

**Figure 4.** Upper panels: similar to Fig. 3c, but for (a): the Schmidt-Appleman criterion (SAC) fulfilled dataset of contrail and natural cirrus (median: $N_{ice}$ = 0.04 cm$^{-3}$ and $R_{ice}$ = 20.6 µm); (b) the SAC unfulfilled dataset of natural cirrus (median: $N_{ice}$ = 0.018 cm$^{-3}$ and $R_{ice}$ = 42.4 µm); (c): the difference of data points between a and b in single $N_{ice}$–$R_{ice}$ bins. Lower panels: similar to Fig. 3d, but for (d): the dataset of contrail or natural cirrus fulfilling SAC; (e): the natural cirrus not fulfilling the SAC; (f): the difference of data points between d and e in single $RH_{ice}$–$T_{amb}$ bins.

### 3.2.1.2 $RH_{ice}$ in the cirrus fulfilling SAC and in natural cirrus

The other pronounced differences between the SAC$^+$ and SAC$^-$ groups are the most frequently appearing $RH_{ice}$ and the respective temperature ranges, as shown in the lower panel of Fig. 4. The highest occurrence frequencies of $RH_{ice}$ in the SAC$_+$ group concentrate at slight ice subsaturation at ~ 90% $RH_{ice}$ in the $T_{amb}$ range of 207–218 K. The slight ice-subsaturation feature is associated with younger contrails with high $N_{ice}$ and small $R_{ice}$, as can been seen from Fig. S2b (see Supplement), where the $N_{ice}$–$R_{ice}$ relation is shown color-coded with $RH_{ice}$ for the SAC$^+$ group in the CA range (see Sect. 2.3.2). In the SAC$^-$ dataset, the highest frequencies of $RH_{ice}$ cluster around 100% at 10 K warmer temperatures and spread over the $N_{ice}$–$R_{ice}$ parameter space (Fig. S2c). The warm temperature range, which is already shown in Fig. 2, reflects the fact that colder temperatures are needed to fulfil SAC because the water saturation pressure at warmer temperatures is almost always so high that the amount of water in the ambient air together with the additional water from aircraft exhaust is insufficient to reach supersaturation with respect to water to form droplets.





In spite of the clear differences in ice particle properties and ambient $RH_{ice}$ conditions resulted from applying SAC (Fig. 4f), ambiguities remain to characterise contrail cirrus and distinguish them from natural cirrus, because the natural cirrus of in situ-origin that have formed at rather low temperatures can show as well the characteristics of medium $N_{ice}$ (0.1 cm$^{-3}$), low IWC and small $R_{ice}$, and would thus be misclassified as contrail cirrus. Additionally, even high $N_{ice}$ accompanied by small $R_{ice}$ can appear in natural cirrus as a result of in situ homogeneously freezing in high updrafts (Krämer et al., 2016).

**3.2.2 Cirrus fulfilling SAC inside and outside the cruising altitude (CA) range**

Here, the SAC$^+$ dataset shown in Fig. 4a and d is split into one group inside the most frequent cruising altitude (CA) range and the other one outside the CA range using the CA pressure boundaries defined in Sect. 2.3.2. The ice cloud properties and $RH_{ice}$ occurrence frequencies related to ambient temperature inside and outside the CA range are presented and discussed below.

**3.2.2.1 Microphysical properties inside and outside the cruising altitude range**

The $N_{ice}$–$R_{ice}$ relation of the cirrus fulfilling the SAC and detected inside the CA range is shown in Fig. 5a, while those outside the CA range are depicted in Fig. 5c. Comparing the $N_{ice}$–$R_{ice}$ relation showing all SAC$^+$ cirrus (Fig. 4a) to those inside and outside the CA range, it becomes visible that the entire group of liquid-origin cirrus ($R_{ice}$ > 30 µm and IWC > 10 ppmv) and a part of the in situ-origin cirrus ($R_{ice}$ < 30 µm and IWC < 10 ppmv) occur outside the CA range, *i.e.*, the cirrus outside the CA range represent a mixture of contrail cirrus, in situ-origin cirrus as well as liquid-origin cirrus, later referred to as cirrus mixture.

Inside the CA range, almost only in situ-origin cirrus are present. The mean $R_{ice}$ and $N_{ice}$ of cirrus particles inside the CA range (Fig. 5a) are approximately 17 µm and 0.21 cm$^{-3}$, corresponding to previous field observations of pure contrail cirrus older than 30 minutes (Schröder et al., 2000; Voigt et al., 2017; Schumann et al., 2017; Chauvigné et al., 2018).

The classification that the cirrus fulfilling SAC and inside the CA range are pure contrail cirrus is confirmed by the validated contrail cirrus, which fulfil SAC and are identified by applying the aircraft plume detection algorithm (Mahnke et al., 2022)

described in Sect. 2.3.3. Since the aerosol and $NO_y$ measurements for both flights on 22 March 2014 were missing and that aircraft plumes detected at $T_{amb}$ > 235 K are screened out, the valid sampling time encountered aircraft exhaust plumes is approximately 0.9 hr, around 3200 cloud samples at 1Hz sampling frequency.

The $N_{ice}$–$R_{ice}$ relation for the validated contrail cirrus is displayed in Fig. 5e. The shape of the overall $N_{ice}$–$R_{ice}$ occurrence frequency distribution for the validated contrail cirrus shows similarity to the pure contrail cirrus differentiated by combining

SAC and the CA range (Fig. 5a), especially when looking at the particle population that contains 50% of the most frequently appearing ice crystals. The median $R_{ice}$ and $N_{ice}$ of the pure contrail cirrus are also close to those of the validated cirrus. In conclusion, combining SAC and the CA range has effectively exposed the differences in the microphysical properties of pure contrail cirrus (SAC$^+$, inside the CA range, Fig. 5a), a cirrus mixture (SAC$^+$, outside the CA range, Fig. 5c) and mostly liquid-origin natural cirrus (SAC$^-$, outside the CA range, Fig. 4b).

Besides, we also inspected the frequency distributions of ice particle sizes, which give further insights into the differences among the cirrus categories and, in addition, confirm the differentiation using the SAC-CA method.



**Figure 5.** $N_{ice}$–$R_{ice}$ (left) and $RH_{ice}$–$T_{amb}$ (right) relations color-coded by normalised occurrence frequency, similar to those in Fig. 4. Top: the contrail cirrus fulfilling the Schmidt-Appleman criterion (SAC) and found inside the cruising altitude range (CA, ambient pressure 200–245 hPa) (median: $N_{ice} = 0.045$ cm$^{-3}$ and $R_{ice} = 16.6$ µm). Middle: the cirrus mixture fulfilling SAC and outside the CA range (in situ- and liquid-origin cirrus) (median: $N_{ice} = 0.038$ cm$^{-3}$ and $R_{ice} = 24.1$ µm). Bottom: contrail cirrus with plume detection applied and fulfil the SAC (median: $N_{ice} = 0.027$ cm$^{-3}$ and $R_{ice} = 21.7$ µm), but the CA range is not considered here.





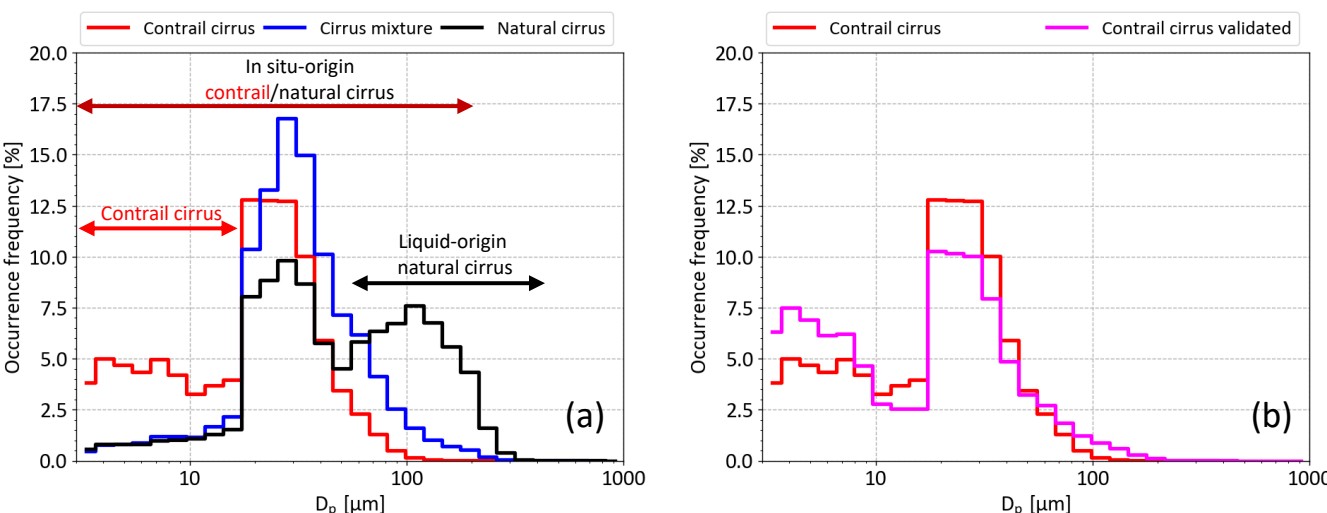

**Figure 6.** (a) Normalised occurrence frequency of ice particle sizes in diameter ($D_p$, unit: µm) in the contrail cirrus (red), cirrus mixture (contrail cirrus, in situ- and liquid-origin natural cirrus, blue), and natural cirrus (black). The ice particle size ranges for contrail cirrus, in situ-origin contrail or natural cirrus, and liquid-origin cirrus are marked by the arrows. (b) Similar to (a) but for contrail cirrus (red) and contrail cirrus satisfying the Schmidt-Appleman criterion (SAC) and validated with the aircraft exhaust plume detection method (magenta).

Figure 6a shows the normalised occurrence frequencies of ice particle sizes in the contrail cirrus (in situ-origin), cirrus mixture (contrail cirrus, in situ- and liquid-origin natural cirrus) and mostly liquid-origin natural cirrus. Three size modes of ice particle sizes can be identified from the frequency distributions of the contrail cirrus and natural cirrus: ice particles in the first size mode — $D_p$ = 3–17 µm (marked by the short red arrow) — appear more frequently inside the CA range and are attributed to pure contrail cirrus; The next size mode — $D_p$ = 3–200 µm (the long red arrow) — is present in contrail cirrus as well as in natural cirrus and is attributed to aged contrails or in situ-origin natural cirrus; the mode $D_p$ = 50–400 µm (the black arrow) originates from liquid-origin cirrus. The maximum sizes represent the largest ice particle size of the particle population including 90% data. Large ice crystals up to about 200–300 µm do appear in contrail cirrus but with a low frequency (see the blue curve in Fig. 6a and also Kübbeler et al., 2011; Voigt et al., 2010). Also, note that 20 µm marks the instrument switches from NIXE-CAS to NIXE-CIPg. This might cause the jump of occurrence frequencies instead of a smooth transition.

The histograms in Fig. 6a show that the contrail (red line) and liquid-origin natural (black line) cirrus are most probably distinguishable in the small ice particle diameter range ($D_p$ < 17 µm) and the larger size range ($D_p$ > 80 µm). Ice crystals between $D_p$ = 17–45 µm occur frequently in the contrail cirrus, but also in the natural cirrus, which makes it difficult to discriminate between the contrail and natural cirrus in this size range. The signature of a larger number of small ice crystals in natural cirrus occurs in the early phase of homogeneous ice nucleation in faster updraughts. However, such events are transient in time and space, and are, therefore, not often found in in situ measurements (Krämer et al., 2020). The contrail cirrus considered here does not contain such homogeneous freezing events. Therefore, ice crystals in the diameter range $D_p$ = 3–17 µm with relatively high occurrence frequencies and the large particles of maximum size of about 200 µm can be attributed to





contrail cirrus, as already noted above. On the contrary, the natural cirrus has the second highest frequency peak in the ice crystals larger than 54 µm with the maximum diameter being about 400 µm. This means that the observed contrail cirrus were formed in situ with the special feature of frequently appearing small ice crystals; the natural cirrus, however, as introduced before, is a mixture of in situ-origin and liquid-origin cirrus. It is to note here that adding an extra constraint of the CA range

to the SAC$^+$ group has greatly minimised the interference of the natural cirrus as well as possibly undistinguishable, deeply aged contrail cirrus of much larger sizes.

The cirrus mixture is interpreted above as a mixture of aged contrail cirrus, mainly in situ-origin cirrus with a small portion of middle-sized liquid-origin cirrus. This is confirmed by the frequency distribution of ice particle sizes shown in Fig. 6a (blue line), where the majority (80% of the total frequencies) of particle diameters are between 17 and 66 µm and the maximum size

is near 300 µm. From this analysis, it is impossible to judge whether the in situ-origin cirrus are aged contrails, which show the same properties as natural cirrus, or whether these cirrus have formed naturally.

Figure 6b shows that the occurrence frequency distribution of ice particle sizes in the contrail cirrus identified using the SAC-CA combination is very similar to that of the validated contrail cirrus aided by the plume detection scheme. In the reference case of the plume-marked contrail cirrus, there are even more smaller ice particles than in the SAC-CA determined contrail

cirrus. Furthermore, the large ice particles occur with very low frequencies not only in the plume-marked contrail cirrus but also in the contrail cirrus constrained by SAC and the CA range. This adds confidence in the discrimination between contrail cirrus and natural cirrus with the SAC-CA combination.

**3.2.2.2 RH$_{ice}$ inside and outside the cruising altitude range**

The RH$_{ice}$–T$_{amb}$ distribution for the contrail cirrus (SAC$^+$, inside the CA range) is shown in Fig. 5b, while Fig. 5d depicts the

cirrus mixture (SAC$^+$, outside the CA range). In comparison to Fig. 4e, where the frequencies of RH$_{ice}$ in the natural cirrus (SAC$^-$) centre around 100% at temperatures above 225 K (also reported in a global RH$_{ice}$ climatology by Krämer et al. (2020), the RH$_{ice}$ inside the contrail cirrus (Fig. 5b) distributes most frequently around 90% and appears almost exclusively in the temperature range T$_{amb}$ = 207–218 K. As mentioned in Sect. 3.2.1, this subsaturation feature is associated with high N$_{ice}$ and small R$_{ice}$ inside the CA range (Fig. 5a), namely the contrail cirrus, as discussed in previous subsections and shown in the N$_{ice}$–

R$_{ice}$ relations color-coded with RH$_{ice}$ in Fig. S2 (see Supplement). Compared to the RH$_{ice}$ distribution in the contrail cirrus (Fig. 5b) and natural cirrus (Fig. 4e), the RH$_{ice}$ frequencies in the cirrus mixture (in Fig. 5d) are more broadly distributed around 100% RH$_{ice}$ between 204 and 229 K, yet with slightly higher frequencies between 80 and 100% at T$_{amb}$ = 207–218 K, similar to the contrail cirrus. The high RH$_{ice}$ values up to 140% in the cirrus mixture are closely related to the in situ-origin cirrus (N$_{ice}$ < 0.1 cm$^{-3}$ and R$_{ice}$ < 30 µm) and liquid-origin cirrus (IWC > 10 ppmv, R$_{ice}$ > 50 µm), as seen from Fig. S2d.

Figure 5f shows that RH$_{ice}$ in the validated contrail cirrus falls mostly below ice saturation in the temperature range of 208–228 K, consistent with the ice-subsaturation feature in the pure contrail cirrus (Fig. 5b). We consider the agreement as verification of the method for separating contrail cirrus from natural cirrus by using only SAC and the CA range.


### 3.3 In-cloud ice sub- and supersaturation — comparisons and causes

In Fig. 7a, we present the $RH_{ice}$ occurrence frequencies of the different cirrus types distinguished using the criteria of SAC,

CA and plume detection: the contrail cirrus, validated contrail cirrus (identified using the plume detection method) and natural

cirrus. $RH_{ice}$ in the contrail and the validated contrail cirrus peaks at 90% $RH_{ice}$, *i.e.*, in slight ice subsaturation, with much

higher occurrence frequency than at 100% $RH_{ice}$. Furthermore, the $RH_{ice}$ distribution in the contrail cirrus tilts to the left —

lower ice subsaturation (80% $RH_{ice}$), while the distribution of $RH_{ice}$ in the natural cirrus has a heavier weight in the right part

of the peaks — more towards ice supersaturation (110%).

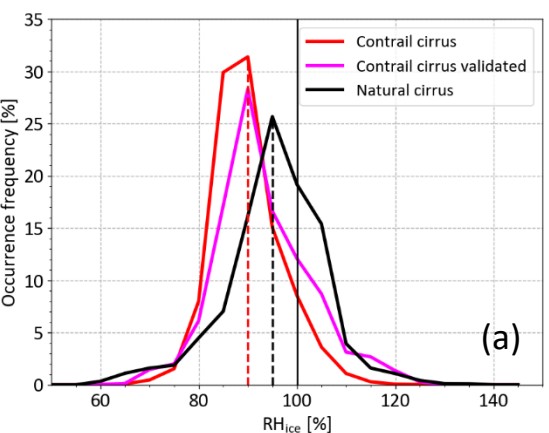 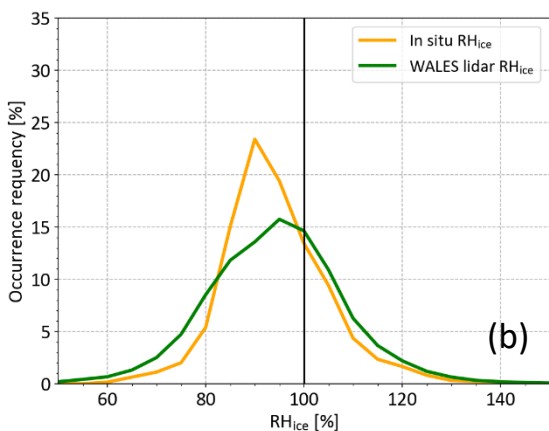


**Figure 7.** (a) Normalised $RH_{ice}$ occurrence frequency distributions in 5% $RH_{ice}$ bin width for the contrail cirrus identified using the combination of the Schmidt-Appleman criterion (SAC) and the cruising altitude range (CA) (red), contrail cirrus validated by the plume detection algorithm (magenta) and natural cirrus not fulfilling SAC and located outside the CA range (black). The most frequently occurring $RH_{ice}$ in the contrail cirrus and natural cirrus are marked by the red and black dashed lines, respectively. (b) Normalised $RH_{ice}$ occurrence

frequency distributions for all cirrus measured in situ (orange) and by the lidar WALES (green).

The in-cloud $RH_{ice}$ occurrence frequency distribution from the WALES lidar observations (see Sect. 2.1) is plotted in Fig. 7b

in comparison to the in situ $RH_{ice}$ of all cirrus. The $RH_{ice}$ distribution of all in situ measured cirrus (the orange curve) peaks at

90% $RH_{ice}$ with an occurrence freqeuncy of ~ 22%, nearly overlapped with the distribution of the lidar $RH_{ice}$ shown in green,

which is broader with a blunt peak around 95% $RH_{ice}$ (~ 15% of occurrence frequency). The lidar $RH_{ice}$ of mixed contrail and

natural cirrus spans from approximately 80–110% at the full width half maximumof the peak, the same $RH_{ice}$ range observed

in most in situ measurements. Despite the different $T_{amb}$ sources used for the in situ $RH_{ice}$ (from BAHAMAS) and lidar $RH_{ice}$

(from ECMWF) calculations, the $RH_{ice}$ distributions related to temperature of in situ (see Fig. 3d) and remote sensing

measurements (see Fig. 8) in the same environment exhibit a consistent view of $RH_{ice}$ occurrence frequencies in the cirrus

clouds in Central Europe and the Northeast Atlantic flight corridor in Spring 2014. The subsaturation feature of cirrus is also

evident in the temperature dependence of the lidar $RH_{ice}$ at cold temperatures between ~ 215–220 K, while at warmer




temperatures above 220 K, $RH_{ice}$ centres at around 100%. The good agreement between the independent in situ and lidar $RH_{ice}$

measurements gives confidence in the assignment of the slight ice-subsaturation feature to contrail cirrus.

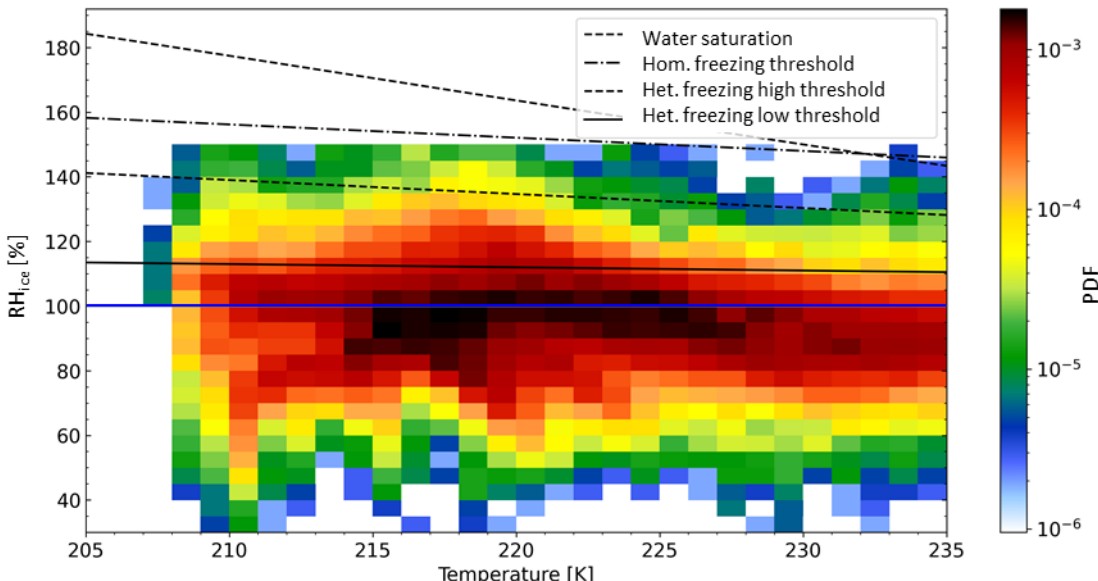

**Figure 8.** Probability distribution of in-cloud $RH_{ice}$ measured by the lidar WALES below 235 K in dependence on ECMWF model temperature. Only the cloud particles producing a back-scattering ratio greater than 3 and depolarisation greater than 20% included in the plot. The water saturation (Murphy and Koop, 2005), homogeneous freezing threshold (Koop et al., 2000), and heterogeneous freezing high (mineral dust as ice nucleating particles) and low (coated soot) thresholds (Krämer et al., 2016) are added in the figure.

However, because of the strong dependence of $RH_{ice}$ on the $T_{amb}$, we discuss here the robustness of the observed subsaturation

feature on changes in temperature. More specifically, the effect of a negative temperature bias om the $RH_{ice}$ distribution is

tested because lower temperatures enhance $RH_{ice}$. The change in $RH_{ice}$ frequency distribution in all cirrus clouds by lowering

the ambient temperature by 0.5 K is shown in Fig. S3 (see Supplement). The peak of the $RH_{ice}$ frequency distribution shifts

from 90% to 95%, so the slight ice-subsaturation feature is still visible in the contrail cirrus above Central Europe and the

Northeast Atlantic region in Spring 2014.

Since finding contrail cirrus in an ice-subsaturated environment may seem surprising, we discuss possible reasons for this in

the following. During the campaign phase in Spring 2014, the background atmosphere in the investigated region was relatively

calm with slow vertical velocities mostly below 0.2 m/s in frontal systems and warm conveyor belts. Why contrail cirrus were

sampled in slight ice-subsaturation can be assumed from two perspectives:

1.  Contrails could have formed in slight ice-supersaturation in pre-existing thin/subvisible cirrus, which have been

    formed heterogeneously and of which the ice particles have grown to large sizes (Kübbeler et al., 2011). Marjani et

    al. (2022) have revealed from satellite retrievals that the perturbation of aircraft on cirrus ice number concentrations

    is located 300–540 metres beneath the flight tracks, right where the primary aircraft vortex descends after formation.





The temperature increase accompanying this descent causes $RH_{ice}$ to decrease to ice-subsaturation accompanied, which was observed for instance by Gayet et al. (2012; Fig. 3h). In such cases, the occurring cirrus would be contrail

cirrus embedded in already existing, possibly subvisible natural cirrus.

2. Contrails could also have formed in slightly ice-subsaturated to slightly supersaturated environments where natural cirrus could not emerge because the threshold humidity for heterogeneous freezing is not reached. But water vapour in the environment together with that emitted from aircraft is sufficient to surpass water saturation and form a contrail in the hot and moist aircraft exhaust. Mixing of the ambient air together with the descend to lower altitudes as

described in (1) would also place the contrail cirrus in a subsaturated environment. The question here is if the ice crystals can grow to the observed sizes of maximum ~ 200 μm during their time in supersaturation. However, this would be the classic case of a blue sky without any cirrus cloud, which turns into a grey sky covered with contrail cirrus in the presence of air traffic.

The next obvious question regarding how long the contrail cirrus can persist in a slightly ice-subsaturated environment will

be discussed in Section 4.

**3.4 Survey of cirrus and contrail cirrus characteristics**

The 10th, 25th, 50th, 75th and 90th percentiles of the different characteristics of the cirrus types, detected over Central Europe in Spring 2014 and separated by the combined analysis of SAC, CA and the plume detection scheme (Section 2.3), are summarised in Table 1. The median $N_{ice}$, $R_{ice}$, IWC and $RH_{ice}$ in the contrail cirrus constrained by SAC and the CA range as

well as those contrail cirrus identified using the aircraft exhaust detection method are correspondent, giving confidence in the new, statistically based contrail–cirrus separation method.

The parameters determining the probable origin, optical property and evolution state of the clouds are the median IWC (Sect. 1), extinction coefficient (Ext) and RHice. Ext is calculated from the empirical formulation in Gayet et al. (2004): $D_{eff} = A \times IWC/Ext$, where the effective diameter $D_{eff}$ is in μm, $IWC$ is in g/m$^3$ and $A = 3000$ mm$^3$/g. The $D_{eff}$ is converted from

the mass mean radius $R_{ice}$, assuming a ratio of $0.7 \pm 0.3$ between $R_{ice}$ and the effective radius $R_{eff}$ of ice particles in contrails and contrail cirrus based on Schumann et al. Schumann et al. (2011). The uncertainty of the Ext is approximately $\pm 43\%$. For contrail cirrus, found in the temperature range 207–218 K, these are IWC = 3.5 ppmv, Ext = ~ 0.056 km$^{-1}$ and $RH_{ice}$ = 88.8%, thus classifying them as in situ-origin, optically thin, evaporating cirrus clouds (Fig. 11, left panel). The warm, natural cirrus (temperature range 225–235 K) are mostly liquid-origin, thick, persisting cirrus (Fig. 11, right panel) cirrus of exhibiting a

higher median IWC (21.7 ppmv), a larger Ext = ~ 0.137 km$^{-1}$ and $RH_{ice}$ = 95.9% close to saturation. The cirrus mixture, with contrails embedded within in situ- and liquid-origin cirrus (Fig. 11, middle panel), are in the intermediate temperature range of 218 to 225 K. Their median properties are IWC = 2.1 ppmv, Ext = ~ 0.096 km$^{-1}$ and $RH_{ice}$ = 94.3%; however, no clear assignment is made here due to their mixed nature.






**Table 1.** Percentiles of ice number concentration $N_{ice}$, mass mean radius $R_{ice}$, ice water content IWC, relative humidity with respect to ice $RH_{ice}$ and Extinction coefficient Ext in the contrail cirrus, the contrail cirrus validated with the aircraft plume detection algorithm, the cirrus mixture and the natural cirrus over Central Europe and the Northeast Atlantic region in Spring 2014.

| Cirrus categories | | Contrail cirrus | Contrail cirrus | Cirrus mixture | Natural cirrus |
|---|---|---|---|---|---|
| SAC[1] | | + | + | + | − |
| CA[2] | | + | + | − | − |
| Plume detection[3] | | NA | A | NA | NA |
| Temperature range | | 207–218 K | 208–217 K | < 207 K, 218–225 K | 225–235 K |
| Data points (1 Hz) | | 14454 | 1270 | 25791 | 12691 |
| $N_{ice}$ [cm$^{-3}$] ($D_p$ > 3 μm) | 10. perc | 0.006 | 0.014 | 0.003 | 0.001 |
| | 25. perc | 0.018 | 0.024 | 0.011 | 0.005 |
| | **50. perc** | **0.045** | **0.041** | **0.038** | **0.018** |
| | 75. perc | 0.135 | 0.131 | 0.112 | 0.069 |
| | 90. perc | 0.454 | 0.642 | 0.309 | 0.170 |
| $R_{ice}$ [μm] | 10. perc | 7.8 | 6.7 | 14.8 | 24.5 |
| | 25. perc | 12.3 | 11.6 | 18.3 | 32.4 |
| | **50. perc** | **16.6** | **17.8** | **24.1** | **42.4** |
| | 75. perc | 20.9 | 21.5 | 37.8 | 55.2 |
| | 90. perc | 24.9 | 24.6 | 52.7 | 65.3 |
| IWC [ppmv] | 10. perc | 0.5 | 0.8 | 0.5 | 1.0 |
| | 25. perc | 1.4 | 2.1 | 2.4 | 6.3 |
| | **50. perc** | **3.5** | **4.4** | **8.3** | **21.7** |
| | 75. perc | 6.4 | 6.6 | 44.7 | 58.1 |
| | 90. perc | 10.9 | 10.0 | 95.4 | 105.0 |
| IWC [mg/m$^3$] | 10. perc | 0.1 | 0.2 | 0.1 | 0.3 |
| | 25. perc | 0.3 | 0.5 | 0.6 | 1.8 |
| | **50. perc** | **0.8** | 1.0 | **2.1** | **6.1** |
| | 75. perc | 1.5 | 1.6 | 13.1 | 16.7 |
| | 90. perc | 2.6 | 2.4 | 28.8 | 31.6 |
| $RH_{ice}$ [%] | 10. perc | 82.5 | 80.5 | 83.4 | 83.1 |
| | 25. perc | 85.5 | 85.7 | 88.9 | 90.5 |
| | **50. perc** | **88.8** | **88.9** | **94.3** | **95.9** |
| | 75. perc | 93.4 | 92.7 | 102.9 | 101.8 |
| | 90. perc | 99.6 | 98.9 | 112.1 | 106.2 |
| Ext[4] [km$^{-1}$] | 10. perc | 0.008 | 0.015 | 0.006 | 0.0109 |
| | 25. perc | 0.023 | 0.036 | 0.031 | 0.045 |
| | **50. perc** | **0.056** | **0.061** | **0.096** | **0.137** |
| | 75. perc | 0.109 | 0.102 | 0.351 | 0.414 |
| | 90. perc | 0.184 | 0.191 | 0.888 | 0.841 |

[1]SAC: the Schmidt-Appleman criterion, "+" fulfilling SAC, "−" not fulfilling SAC.
[2]CA: the cruising altitude range, "+" inside CA, "−" outside CA.
[3]Plume detection: the plume detection algorithm, "A" applied, "NA" not applied.
[4]Ext: the extinction coefficient. It's calculated after the equation (3) in Gayet et al. (2004); See the text for details.





## 4 Persistent cirrus in slight ice-subsaturation – potential influence on aviation's climate impact

The slight ice-subsaturation feature of the contrail cirrus observed over Central Europe in Spring 2014 agrees with the occurrence of contrails in ice subsaturated atmosphere that was observed during the CONCERT campaign (Kübbeler et al., 2011; Voigt et al., 2010; Gayet et al., 2012), although contrail cirrus crystals sampled during ML-CIRRUS were much older (Schumann et al., 2017; Voigt et al., 2017) than those young contrails at the age of a few minutes detected during the CONCERT campaign (Voigt et al., 2010; Chauvigné et al., 2018). A comprehensive compilation of contrails and

contrail cirrus measurements from a series of research aircraft campaigns confirmed that the occurrence in slight subsaturation with respect to ice is a pronounced characteristic of contrail cirrus (Schumann et al., 2017). Furthermore, Petzold et al. (2017) reported the observation of contrail cirrus in slight ice subsaturation in the North Atlantic flight corridor using $RH_{ice}$ measurements aboard the passenger aircraft in the IAGOS research infrastructure. Concluding, contrail cirrus occurrence in slight ice subsaturation is not an uncommon feature.

Whether the contrail cirrus existing in slight ice subsaturation might affect the radiative forcing is connected to the lifetime of ice crystals in such environment. Therefore, we investigated the lifetime of cirrus ice particles of the size and concentration identified for contrail cirrus in slight ice subsaturation using the SAC-CA combination. A scenario of cirrus cloud particles formed at $T_{amb} < 210$ K was simulated using the detailed microphysical box model MAID (Model for Aerosol and Ice Dynamics) (Bunz et al., 2008; Rolf et al., 2012; Krämer et al., 2016). The simulation was initialised with a water amount of

90% $RH_{ice}$ ($T_{amb}$) and INPs at 0.1 $cm^{-3}$. The adiabatic cooling or warming rate at vertical wind speeds of 10 cm $s^{-1}$ was added to present the constant updraughts or downdraughts in the atmosphere. Ice crystal sublimation and sedimentation processes were not considered in the simulation. The formation and evolution of cirrus particles in the simulated scenario are plotted in Fig. 9. Although contrail formation relevant processes are not included in the model, $N_{ice} = 0.2$ $cm^{-3}$ and $R_{ice} = 17$ µm at the time when the temperature reaches the plateau (Fig. 9 top panel) before the starting of the warming phase are comparable to

the contrail cirrus observed during ML-CIRRUS flights (Fig. 5a, within the 50% contour). As the warming procedure reduces $RH_{ice}$ to below ice saturation, cirrus ice particles of similar properties to the observed contrail cirrus gradually diminish, which takes approximately 4 h for the ice particles to sublimate until $RH_{ice}$ declines to below 80%. This implies that contrail cirrus existing in a slightly ice-subsaturated environment could survive for a long-time scale, during which they might also alter Earth's radiation budget, similar to the persistent contrails formed in ISSRs.

Whether slightly ice-subsaturated regions are relevant for the influence of contrail cirrus on climate depends on the frequency of the occurrence of such regions. For this reason, we assessed the changes in air masses for a further potential radiative impact of contrail cirrus when lowering the threshold of contrail persistence in $RH_{ice}$ from 100% to 90%. Figure 10 summarises the air mass percentages under different $RH_{ice}$ thresholds averaged over the North Atlantic and Europe for the IAGOS-MOZAIC observational period from 1995 to 2010 (see Petzold et al., 2020 for details). The air mass

percentage of $RH_{ice} \geq 90\%$ for the Europe and North Atlantic flight corridor is approximately 43%, increased by more than 10% in comparison to the air mass percentage of $RH_{ice} \geq 100\%$. Whether this finding might lead to a larger impact




of aviation on the climate is unclear, particularly when the recent results on the relevance of strong warming contrails, the so-called big hits (Teoh et al., 2020b; Gierens et al., 2020) are considered.

The climate impact of aviation-induced cirrus clouds by today's knowledge is predominantly linked to contrail cirrus in ISSRs.
So is the current recommendation for diverting the aircraft to avoid the formation of contrails with strong radiative forcing potential (Teoh et al., 2020a; 2020b). It's unclear how the existence of contrail cirrus in slightly ice-subsaturated regions could influence the assessment of the climate impact of contrails and contrail cirrus. However, as we have shown in our analysis, contrail cirrus do survive several hours in such slight ice-subsaturation (90% $RH_{ice}$), and there might be a non-negligible increase in contrail cirrus coverage if considering the existence of contrail cirrus in $RH_{ice} \geq 90\%$. Thus, we recommend
considering the slightly ice-subsaturated regions for the benefit of safe contrail avoidance during air traffic management and a reliable estimation of the radiative forcing of aviation-induce cloudiness.

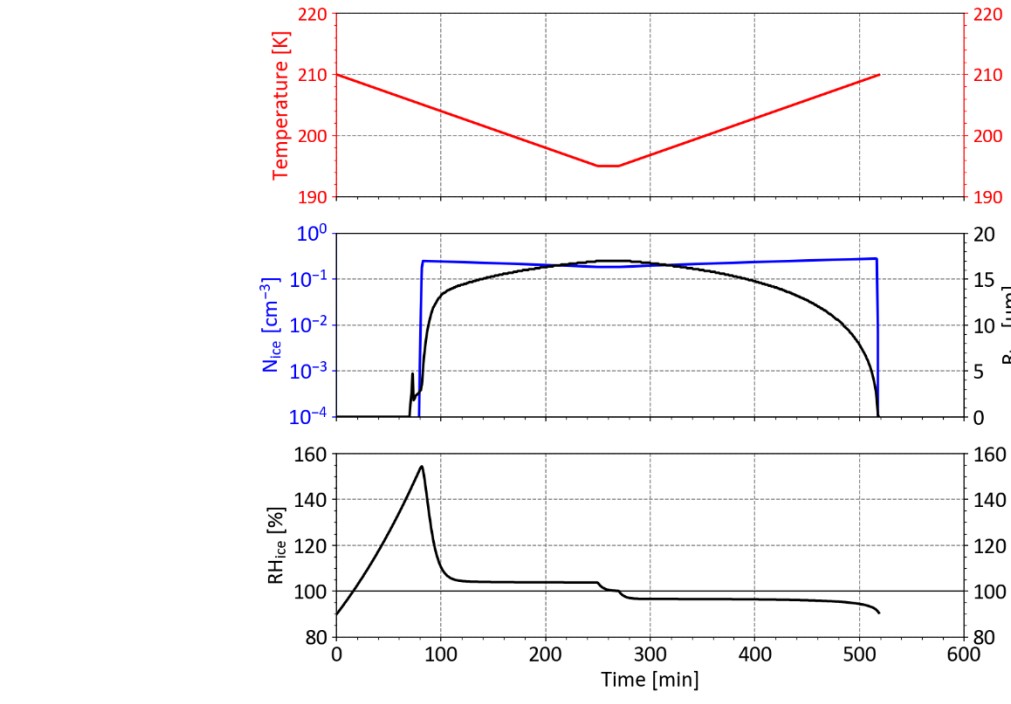

**Figure 9.** Simulated evolution of cirrus cloud particles initialised at 210 K and 90% $RH_{ice}$ in dependence on the simulation time in minutes. Top: temperature of air parcels (unit: K). Middle: ice particle number concentration $N_{ice}$ (cm$^{-3}$, blue) and mass radius mean $R_{ice}$ (µm, black) of cirrus crystals. Bottom: relative humidity with respect to ice $RH_{ice}$ (%).





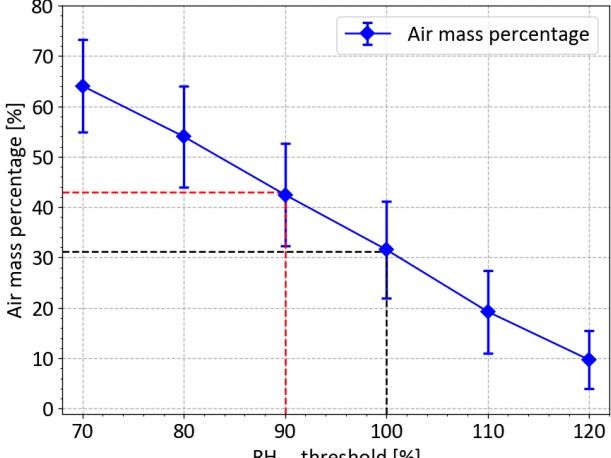

**Figure 10.** Percentage of air masses in dependence on RH$_{ice}$ thresholds in the upper troposphere, averaged over the North Atlantic (65°–5° W) and Europe (5° W–30° E) regions for the MOZAIC period from 1995 to 2010. The percentages of air masses above 90% and 100% RH$_{ice}$ thresholds are labelled with the red and black dashed lines, respectively.

## 5 Summary

Fresh contrails can be easily identified owing to their brightness and linear shape. Apart from that, contrail cirrus, especially those aged into thin cirrus layer, are difficult to be separated from natural cirrus to tackle aviation-induced climate impact. In this work, a new approach to filter out contrail cirrus from natural cirrus is developed: we combined the Schmidt-Appleman criterion (SAC) — the fundamental thermodynamical approach to predict contrail formation — with a new aircraft exhaust plume detection algorithm to statistically discriminate between the contrail and natural cirrus measured above Central Europe during the ML-CIRRUS research aircraft campaign in Spring 2014.

Cloud particles matching SAC are presumably attributed to contrail cirrus, while those missing SAC are treated as natural cirrus. Comparatively young contrail cirrus were encountered most frequently in the cruising altitude (CA) with ambient pressure ranging from 200 to 245 hPa (ambient temperature range 207–225 K). Figure 11 summarises the microphysical and optical properties of the contrail and natural cirrus observed during the ML-CIRRUS. It shows that the microphysical and optical properties of the contrail (Fig. 11, left) and natural cirrus (Fig. 11, right) differ markedly with the contrail cirrus occurring in a much higher median number density N$_{ice}$ = 0.045 cm$^{-3}$ accompanied by a smaller mass mean radius R$_{ice}$ = 16.6 μm and mostly ice water content IWC < 10 ppmv, compared to N$_{ice}$ = 0.018 cm$^{-3}$, R$_{ice}$ = 42.4 μm and IWC frequently above 10 ppmv in the natural cirrus. The relatively low extinction coefficients of the contrail cirrus (median 0.056 km$^{-1}$ compared to 0.137 km$^{-1}$ in the natural liquid-origin cirrus) reveal that the observed contrail cirrus clouds were rather thin, indicating aged contrail cirrus particles. Altogether, the contrail cirrus sampled inside CA share the characteristics of in situ-origin cirrus, in





contrast to the large and optically thicker natural cirrus of liquid-origin. Cirrus clouds outside CA (Fig. 11, middle) are a complex of in situ and liquid-origin cirrus with contrails embedded.

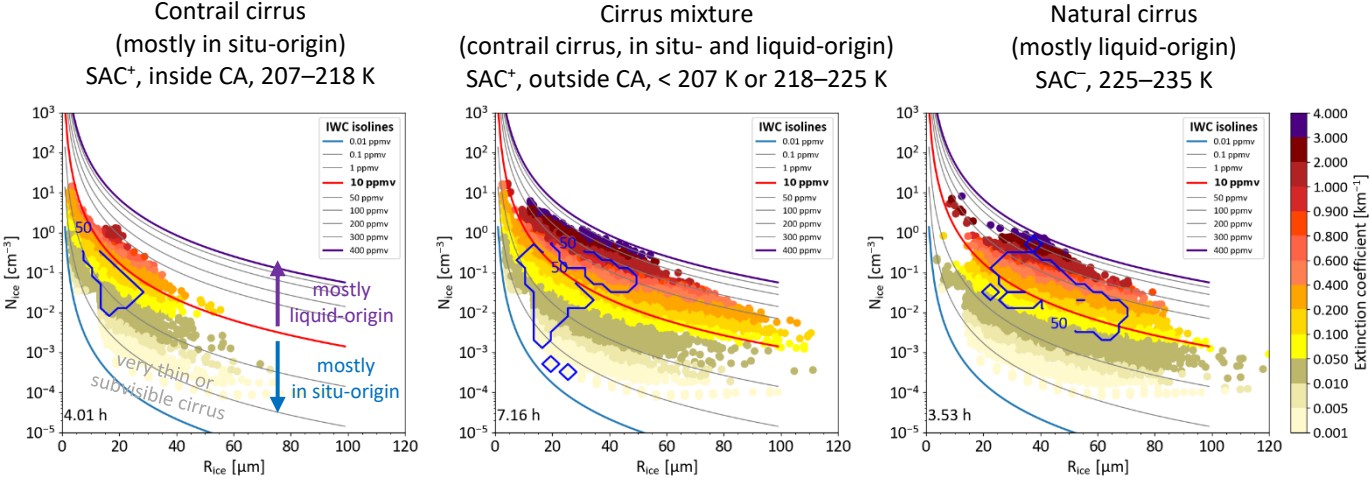

**Figure 11.** Ice crystal number concentration $N_{ice}$ vs. mass mean radius $R_{ice}$ colour coded with extinction coefficient Ext (unit: km$^{-1}$) of ice particles. Ice water content IWC isolines in relation to $N_{ice}$ and $R_{ice}$ are also plotted. Left: the contrail cirrus that fulfil the Schmidt-Appleman criterion (SAC) inside the cruising altitude (CA; ambient pressure 200–245 hPa, ambient temperature 207–218 K), also identified as in situ-origin cirrus; median: $N_{ice}$ = 0.045 cm$^{-3}$, $R_{ice}$ = 16.6 µm, IWC = 3.5 ppmv, Ext = ~ 0.056 km$^{-1}$. Middle: the cirrus mixture (fulfilling SAC and outside the CA range), which is a mixture of contrail cirrus, in situ- and liquid-origin natural cirrus; median: $N_{ice}$ = 0.038 cm$^{-3}$, $R_{ice}$ = 24.1 µm, IWC = 8.3 ppmv, Ext = ~ 0.096 km$^{-1}$. Right: the natural cirrus of liquid-origin (not fulfilling SAC and below the CA range); median: $N_{ice}$ = 0.018 cm$^{-3}$, $R_{ice}$ = 42.4 µm, IWC = 21.7 ppmv, Ext = ~ 0.137 km$^{-1}$. The 50% (blue) contour enclose 50% the most frequently occurring ice crystals. The total sampling hours of each cirrus dataset at 1Hz sampling frequency are inserted in the lower left corners of the figures.

An important finding of this study is that the highest probability of $RH_{ice}$ in the contrail cirrus occurs in slight ice-subsaturation, centring at around 90% $RH_{ice}$, concurring with previous studies based on a smaller dataset of in situ measurements. The $RH_{ice}$ distribution in the natural cirrus agrees with the worldwide climatology compiled by Krämer et al. (2020; Fig. 7), which centres around ice saturation at T > ~ 200 K. The existence of contrail cirrus in slightly ice subsaturated environments seems to be surprising from the perspective of thermodynamic equilibrium, but Krämer et al. (2020) and Jensen et al. (2001) also reported natural cirrus under subsaturated conditions. As predicted by a cirrus scenario simulated with the box model MAID, contrail cirrus can persist up to over 4 h in around 90% $RH_{ice}$ environment, which means that contrail cirrus likely plays a role in the overall contrail radiative feedback provided that they appear frequently. An estimation of the air mass percentage with $RH_{ice}$ ≥ 90% in air traffic cruise regions, based on 15 years of MOZAIC $RH_{ice}$ measurements, shows an increase by approximately 10% in comparison to the air mass percentage in the ISSRs. This suggests that we need to lower the $RH_{ice}$ threshold to achieve the efficacy of contrail avoidance by rerouting aircraft. We also call for deeper investigations into the spatial coverage and optical depth of the contrail cirrus in slight ice-subsaturated environments as well as the associated microphysical processes to predicate their climate effect robustly. In turn, this will also facilitate the mitigation of aviation's climate impact by reducing the occurrence of contrails and contrail cirrus.



**Data availability.** The data that support the findings of this study are available upon request from the authors. The IAGOS
data are available through the IAGOS data portal at https://doi.org/10.25326/20 (Boulanger, 2021).

**Author contributions.** YL and MK designed the study. YL carried it out and prepared the manuscript with contributions from
all co-authors. CM and UB developed and applied the plume detection algorithm to identify cirrus particles influenced by
aircraft exhaust. SR and AP analysed the pressure levels of IAGOS-MOZAIC flights over the North Atlantic and Europe and
provided air mass fractions at different $RH_{ice}$ thresholds in this region observed during IAGOS-MOZAIC period. NS prepared
the PSDs data for calculating the frequency of ice particle sizes. MK performed the cirrus life cycle simulation. GD and SG
provided the $RH_{ice}$ data from WALES. CV and US coordinated the ML-CIRRUS campaign.

**Competing interests.** Two of the co-authors are members of the editorial board of Atmospheric Chemistry and Physics.

**Acknowledgement.** We would like to thank Martin Zöger (DLR, Germany) for providing BAHAMAS and SHARC data from
ML-CIRRUS, Armin Afchine (FZJ, Germany) for NIXE data, Daniel Sauer (DLR, Germany) for aerosol data and Helmut
Ziereis (DLR, Germany) for providing $NO_y$ data. We are grateful to Klaus Gierens (DLR, Germany) for helpful discussion.
MOZAIC/IAGOS data are created with support from the European Commission, national agencies in Germany (BMBF),
France (MESR), and the UK (NERC), and the IAGOS member institutions (http://www.iagos.org/partners). The participating
airlines (Deutsche Lufthansa, Air France, China Airlines, Iberia, Cathay Pacific, Hawaiian Airlines, Air Namibia, Sabena,
Austrian) supported IAGOS by carrying the measurement equipment free of charge since 1994. The data are available at
https://doi.org/10.25326/20 thanks to additional support from AERIS. MK thanks JGU Mainz for support as a GFK fellow.
CV was funded by the Helmholtz excellence program (grant number W2/W3-060) and by the German Research Foundation
DFG in SPP 1294 HALO under contract VO 1504/7-1 and the TRR 301 – Project-ID 428312742.

**Financial support.** This research has been supported the EU Horizon 2020 project Advancing the Science for Aviation and
ClimAte (ACACIA grant No. 875036 and the German Ministry for Education and Research (BMBF grant No. 01LK1301A).

The article processing charges for this open-access publication were covered by a Research Centre of the Helmholtz
Association.



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
