# Peer review of "Upper tropospheric slightly ice-subsaturated regions: Frequency of occurrence and statistical evidence for the appearance of contrail cirrus"

_Atmospheric Chemistry and Physics, 2022_

## Referee Comment (RC2)

**Review of "Upper tropospheric slightly ice-subsaturated regions: Frequency of occurrence and statistical evidence for the appearance of contrail cirrus" by Li et al. (2022)**

**Overview**

This study is focused on the analysis of in-situ microphysical measurements of contrail cirrus and natural cirrus clouds based on the data collected during the ML-CIRRUS campaign. An important component of this work is the attempt to segregate contrail cirrus embedded in natural cirrus. One of the major outcomes of this study is the statistics of $RH_{ice}$ which are suggestive of a bias in the average humidity in contrail and natural cirrus clouds towards undersaturaion, ranging, on average, from approximately 4% to 12%. The paper is well organized and undoubtedly deserves publication.

**Recommendation**: The paper should be published in ACP after addressing the comments indicated below.

**Comments**

1. Methodology: Identification of contrails embedded in cirrus and contrail cirrus clouds, within the P and T ranges, predetermined by CA, was based on the analysis of (a) the Schmidt-Appleman criterion (SAC) and (b) measurements of engine combustion products, aerosols and $NO_y$ (aircraft plume detection). A potential caveat of this approach is that $NO_y$ is a passive tracer, whereas cloud particles are an active cloud admixture in the atmosphere with a different response to the force of gravity and turbulent motions. As a result, at some point the contrail ice particles may become spatially separated from the plume and/or the plume may become spatially associated with particles formed in natural cirrus clouds. An explanation regarding this matter would clarify the limitations of the applied methodology. Specifically, what is the maximum age of contrail cirrus clouds when this method can be applied?

2. As indicated in Table 1, the plume detection was only applied to approximately 2% of the collected data set. This brings up a question about the statistical significance of this data subset compared to data set with the SAC only criterion applied. It also would be relevant to state upfront in section 2.3.3 that the plume detection was applied only to a small fraction of the collected data, rather than having the reader figure it out after analysis of the data statistics in Table 1, at the end of the paper.

3. Airborne measurements of $RH_{ice}$ at temperatures below -50C are known to be of great challenge. It appears that the accuracy of the $RH_{ice}$ measurement required for the main outcomes of this paper should be of the order of 1%. Even though $RH_{ice}$ is one of the key parameters in this study, there are no discussions of the accuracy of measurements, inflight checks of the performance of humidity probes, etc. A brief discussion of this topic would be highly relevant in this paper, and it facilitate its reading rather than surfing through references. In this regard, I am wondering if you attempted inflight calibrations of water vapor probes in liquid clouds based on the methodology proposed in Korolev and Isaac (2006, JAS, https://doi.org/10.1175/JAS3784.1)?

4. Section 4. I found the discussion around Figure 9 a bit misleading. The diagram in Figure 9 shows changes of T, R_{ice}, and S_{ice} in an adiabatically ascending and then ascending parcel. The supersaturation in the vertically moving parcel will set to its quasi-steady value $S_{qs} = \frac{au_z}{N_{ice}\bar{r}_{ice}}$ at time $t > 3\tau_{ph}$ , where $\tau_{ph}$ is the time of phase relaxation (see Korolev and Mazin, 2003, JAS, https://doi.org/10.1175/1520-0469(2003)060%3C2957:SOWVIC%3E2.0.CO;2). The two plateaus with $S_{qs}$>0 and $S_{qs}$<0 for the ascending and descending branches, respectively, are clearly visible in Fig.9. However, the authors consider only the descending branch, where the supersaturation is negative, and use it as an argument to explain the negative bias of RH_{ice} in cirrus clouds. However, in stratiform type clouds, vertical ascending and descending motions are approximately equally probable, and the distribution $F(u_z)$ is typically centered around 0. Keeping this in mind, and that $S_{qs}(u_z) = -S_{qs}(-u_z)$, the spatial averaging of humidity will yield $S \approx 0$.

In addition to the above, it is worth mentioning that complete evaporation of particles in adiabatic parcel will occur at the same level $Z_{ev}$., which depend on initial $IWC$ and the level $Z_0$. (To be strict, the level of complete sublimation depends on $u_z$. However, for the sake of argument, this effect of the condensational inertia can be neglected here.) Therefore, the lifetime of a descending cirrus parcel can be to a first approximation estimated as $t\sim(Z_0 - Z_{ev})/u_z$. Therefore, the estimated longevity of the subsaturated cirrus as 4h is a function of $u_z$ and $IWC(Z_0)$.

Having said the above, I would suggest reconsidering the argumentation in section 4 and the statement about 4h lifetime in the abstract.

5. I attempted a simulation of the response of cirrus at $u_z = 0$ to the subsaturated environment with $RH_{ice}(0) = 90\%$, and the same $N_{ice}$ and $R_{ice}$ as indicated in Section 4. The results are shown in three diagrams to the right. It turned out that the in-cloud air arrives to saturation within ~25min. The red vertical line indicated $\tau_{ph}$ for initial $N_{ice}(0)$ and $R_{ice}(0)$. $\tau_{ph}$ shows a typical time of reaching saturation (usually within $3\tau_{ph}$ ). In this regard, it would be highly beneficial to indicate in Table 1 the time of phase relaxation.

[Figure]

[Figure]

[Figure]

6. IAGOS-MOSAIC data: I believe that the autonomous instruments installed in the commercial passenger aircraft in the frame of IAGOS were not maintained and calibrated with the same depth and frequency as on the HALO research airplane. Even though there are several references in the paper about the IAGOS data quality, it would be helpful to see a few general statements about the accuracy of $RH_{ice}$ measurements.

**Minor comments**

1. Lines 13, 101, 266: It is not clear what the spatial statistics of the sampled clouds is. It is worth indicating the total length of sampled clouds along with the total cloud sampling time 14.7h.

2. Line 141: In the equation for $R_{ice}$ the notations, "$1.e^4$" and "$1.e^{-6}$" are confusing. It should be "$10^4$" and "$10^{-6}$".

3. Section 2.1, Figure 6 and associated text: It would serve to clarify the paper to use the same type of definition of particle size, rather than switching between radius and diameter. Also indicate the definition of $D_p$., i.e., max particle size, average projected size, equivalent volume size, etc.

4. Table 1. I found that IWC ($mg/m^3$) calculated from $N_{ice}$ and $R_{ice}$ based on Eq. on line 141 is systematically lower than those indicated in Table 1. Was IWC ($mg/m^3$) calculated from IWC (ppmv)? A brief explanation in a footnote would be relevant.

5. Figure 6b: The colors of PSDs for 'Contrail cirrus' and 'Contrail cirrus validated' appear to be the same (magenta and red). It is highly recommended to replace one of the colors by e.g., blue, violet, green, black for a better visualization of the curves.

6. Figure 7a: same as in #4.

7. Figure 8: This diagram uses the same type of lines (i.e., dashed and solid) to indicate different curves.

8. Line 651: "rather thin" => "rather optically thin".

Alexei Korolev

---

## Author Response (AR2)

**Authors' response to reviewers**

**Reply to Reviewer #1**

We thank Minghui Diao for taking the time to carefully read through the manuscript and the generally positive comments. Please find below the reviewer's comments in normal text, with our response in blue and changes that has been made in the revised version of the manuscript in red.

**RC1**: In Figure 7, the occurrence frequency of $RH_{ice}$ in natural cirrus peaks at 95%. But the authors described this figure as the $RH_{ice}$ centers at 100%: (line 460) "In comparison to Fig. 4e, where the frequencies of RHice in the natural cirrus (SAC–) centre around 100% at temperatures above 225 K (also reported in a global $RH_{ice}$ climatology by Krämer et al. (2020), …" The reviewer wonders if this suggests that the water vapor measurements or the combination of water vapor and temperature measurements in ML-CIRRUS has a low bias by 5%? The distributions of all in-cloud $RH_{ice}$ for in-situ and remote sensing observations also suggest there may be a low bias for in-situ observations. If this is the case, then the subsaturated conditions for contrail cirrus would be more around 95% instead of 90%. Previously, several studies on US NSF-funded field campaigns analyzed in-situ measurements of RHice for cirrus clouds. They all showed a peak position at 100% for RHi distribution.

Figure 12b in Patnaude, R., M. Diao, X. Liu, S. Chu. Effects of Thermodynamics, Dynamics and Aerosols on Cirrus Clouds Based on In Situ Observations and NCAR CAM6 Model. Atmospheric Physics and Chemistry, 21, 1835–1859, https://doi.org/10.5194/acp-21-1835-2021, 2021

Figure 5 in Diao, M., G.H. Bryan, H. Morrison, and J.B. Jensen, Ice nucleation parameterization and relative humidity distribution in idealized squall line simulations, Journal of the Atmospheric Sciences, 74, 2761–2787, https://doi.org/10.1175/JASD-16-0356.1, 2017.

Figure 4 in Diao, M., M.A. Zondlo, A.J. Heymsfield, L.M. Avallone, M.E. Paige, S.P. Beaton, T. Campos and D.C. Rogers. "Cloud-scale ice supersaturated regions spatially correlate with high water vapor heterogeneities", Atmospheric Chemistry and Physics, 14, 2639-2656, 2014.

The references were inserted in Line 441.

Can the author look more closely into the time series of the flights, and see if there was possible bias in RHi measurements? One possible method is to look at $RH_{liq}$ for warm clouds and they should be very close to 100% liquid saturation. Although this method may not work well if the bias from the instrument is temperature dependent (which you should be able to tell from lab calibrations). Did the SHARC instrument participate in any water vapor intercomparison experiment, or lab comparisons with commercial chilled mirror hygrometer such as RHS system (accuracy +/-1 0.1degC)? Another possible method is to examine typical cirrus clouds sampled in ML-CIRRUS, and especially the ones mixed with ice supersaturated segments. When the ice crystal regions and clear-sky ice supersaturated regions are intermittently observed, it is often that the ice crystal regions show ice saturation or slight ice supersaturation instead of ice

subsaturation. If these segments frequently show ice subsaturation when they are surrounded by clear-sky ice supersaturation, it would be an indicator of possible low bias in $RH_{ice}$.

The uncertainty of water vapor instrument, temperature probe, and the combined $RH_{ice}$ uncertainty from water vapor and temperature should be added in the description around line 125.

**AC1**: The SHARC instrument was deployed on board HALO together with the Fast In-situ Stratospheric Hygrometer (FISH) and the Atmospheric Ionization Mass Spectrometer for water vapor (AIMS) during the ML-CIRRUS campaign. The overall uncertainty of SHARC $H_2O$ measurement is 5% relative and ±1 ppm absolute offset uncertainty (Kaufmann et al., 2018b). The nominal accuracies of the BAHAMAS pressure and $T_{amb}$ measurement are 0.3 hPa and 0.5 K (Mallaun et al., 2015; Giez et al., 2017; Kaufmann et al., 2018b). The overall accuracy of the in-situ $RH_{ice}$ measurements here is between $10 - 20\%$, with the respective uncertainties of the temperature, pressure and water vapour measurements considered (Krämer et al., 2016). The description of instrumental uncertainties and accuracies was added to Line 126 in the revised manuscript.

We looked into the flights focusing on natural cirrus. In general, the $RH_{ice}$ in cirrus clouds is higher than neighbouring clear-sky conditions. The $RH_{ice}$ in cirrus clouds is above ice saturation in flight segments when cirrus regions and clear-sky ice supersaturated regions appear intermittently. This holds true during the contrail-dedicated flights, but the overall $RH_{ice}$ is below ice saturation, with some cases where ice crystals and neighbouring clear-sky conditions are observed in ice supersaturated regions. Besides, intercomparisons for SHARC, FISH (calibration before and after flights) and AIMS (in-flight calibration) $H_2O$ measurements showed very good agreement between instruments (Meyer et al., 2015; Kaufmann et al., 2018b). No strong bias was found in either water vapour instruments or temperature measurements. Also, in the past years, unpublished work of intercomparisons between SHARC, FISH (the high precision hygrometer from Jülich) and other water vapour instruments during field campaigns suggests SHARC is a very robust instrument.

However, the possibility of a small bias in the in situ $RH_{ice}$ dataset due to a positive bias in the Basis Halo Measurement and Sensor System (BAHAMAS) temperature measurement was brought up in Schumann (2021; See page 108). Here, the impact of the positive temperature bias of 0.5 K on the $RH_{ice}$ distribution is addressed in Sect. 3.3 starting from Line 492 and more details can be found in Sect. S3 in the Supplement. With the positive temperature bias considered, the subsaturated conditions for contrail cirrus would peak at 95%, shifting by 5%, and the peak of natural cirrus $RH_{ice}$ distribution would move closer to 100%. In the revised version, we refer to the discussion about the effect of a possibly positive temperature bias already at the beginning of Sect. 3.3.

**RC2**: The reviewer suggests adding an analysis on the distribution of RHi for inside contrail cirrus with respect to the cruising altitude. If the author calculate delta_z or delta_p for each second of flight data with respect to cruising altitude, and plot RHi only for inside contrail cirrus (CA + SAC methods), will the RHi distribution show more ice supersaturation on the higher levels and more subsaturation in the lower levels? This can help verify if these contrails in the sub-saturated conditions happen due to ice crystals sedimenting into lower altitudes with subsaturated conditions, or the contrail ice crystals stay at similar altitudes, but their environmental condition gradually becomes subsaturated.

**AC2**: As the referee suggested, we plot the $RH_{ice}$ distribution for contrail cirrus (SAC+CA) with delta_p ($\Delta p$) calculated for each second of flight data during the contrail-dedicated flights, see Fig. 1a. In addition, the distribution of $RH_{ice}$ in relation to temperature and $\Delta p$ is plotted in Fig. 1b. The ice crystals showing ice supersaturation was appearing more often in the lower part of CA (p > 222.5 hPa in Fig. 1a, $T_{amb}$ > 212 K in Fig. 1b), *i.e.*, the ice subsaturation speared more frequently in the upper CA in spite of some ice-supersaturated air masses. From this point of view, it is still difficult to verify from the $RH_{ice}$-$\Delta p$ relation vs. pressure p (Fig. 1a) or temperature (Fig. 1b) if ice crystals sedimented in subsaturated region or if the air mass gradually became subsaturated.

[Figure]

Figure 1. (a): Ambient pressure vs. delta_p ($\Delta p$), color-coded with $RH_{ice}$ for contrail cirrus (SAC+CA) with $\Delta p$ calculated for each second of flight data during the contrail-dedicated flights. (c): Similar to (a), but for ambient temperature vs. $\Delta p$. (c): Altitude vs. $RH_{ice}$ for contrail cirrus (SAC+CA), color-coded with occurrence frequency. The bin widths for $RH_{ice}$ and altitude are 5% $RH_{ice}$ and 200 m, respectively. The total sampling time of the contrail cirrus satisfying SAC and CA is 3.8 h, added in the figure.

Therefore, we plot the $RH_{ice}$ in 5% bins vs. the flight altitude in 200 m bins color-coded by occurrence frequency, shown in Fig. 1c. Here, we can see that the ice supersaturation (ISS, in total 8.9%) was mostly encountered between $10.8 - 11.1$ km, the lower altitudes of CA range, where ice subsaturation also occurred most frequently, with the second highest frequency in higher altitudes ($11.1 - 11.7$ km). It points out that the air mass gradually became subsaturated. The ice-subsaturation might be related to aged contrails, possibly as a result of cirrus sublimation in the environment that gradually becomes subsaturated due to the

entrainment of cold and dry ambient air. The ice supersaturation seems to be related to very young contrails formed at the early stage of the detrainment of hot and humid aircraft exhaust into cold ambient air, because it was mostly detected in warmer temperature regions (212 – 226 K).

**RC3**: In Figure 3, can the authors add a third row, for $N_{ice}$ versus $R_{ice}$ and $RH_{ice}$ versus temperature (similar to Figure 3 c and d), but categorize the samples into two groups, (1) fulfilling the plume detection criterion or (2) not fulfilling that criterion? It is unclear where the samples fulfilling that plume detection criterion would be distributed, and how they are related to the SAC and CA criteria.

Figure 5 would also benefit from an additional row, illustrating Cirrus: fulfilling SAC, inside CA, and also with restriction to plume detection. The reviewer wonders if applying a third restriction of plume detection criterion to the combined SAC+CA criteria would make a big difference.

**AC3**: This is a good suggestion. We have included an extra section in the supplement in the revised version. In the new version, another section Sect. S4 and Fig. S4 (shown below) were added into the Supplement, explaining the effect of applying the plume detection algorithm on the separation of contrail and natural cirrus using SAC and CA. And the conclusion of Sect. S4 was inserted in Line 388 in the revised manuscript.

The $N_{ice}$-$R_{ice}$ distribution for the cirrus fulfilling and not fulfilling the plume detection criteria are plotted in Fig. S4a and c. Figure S4b and d show the corresponding $RH_{ice}$-$T_{amb}$ relations. Because the plume detection depends greatly on $NO_y$ and aerosol concentrations, and the enhancement signal of the species decays with time, only rather fresh plumes younger than about 4 h can be identified. Therefore, the population of cirrus particles that can be traced back to plumes is rather small, 0.99 h, as shown in Fig. S4a. A large number of the ice crystals found in plume (0.86 h) fulfil SAC (Fig. 5e). Temperature wise, most of the ice particles are found in the CA temperature range (207 – 218 K) with a high occurrence frequency in ice subsaturation, see Fig. S4 b. Most of the cirrus cannot be validated with the plume detection algorithm. They are a mixture of contrail cirrus, in situ- and liquid-origin natural cirrus, spreading in a wide temperature range, see Fig. S4c and d.

The cirrus crystals fulfilling SAC, inside CA and also with restriction to plume detection are shown in Fig. S4e, with their $RH_{ice}$ vs. $T_{amb}$ displayed in Fig. S4f. From Fig. S4e, we can see that the $N_{ice}$-$R_{ice}$ distribution of ice particles would be represented nearly by the 50th percentile in Fig. 5e. Comparing the median $N_{ice}$ and $R_{ice}$ values represented by Fig. 5e and Fig. S4e (which is also listed in Table 1), we can see that the medians in the dataset using SAC, CA and plume detection are closer to but not significantly different from the ones determined using only the combined SAC+CA. Therefore, it does not make a big difference to add the restriction of plume detection to the combined SAC and CA criteria.

[Figure]

Figure S4. $N_{ice}$–$R_{ice}$ relations (left) and $RH_{ice}$-$T_{amb}$ relations (right) color-coded by normalised occurrence frequency, similar to Fig. 3c and d. (a): $N_{ice}$–$R_{ice}$ relation for the ice particles found in aircraft plumes using the plume detection algorithm (median: $N_{ice}$ = 0.027 cm$^{-3}$, $R_{ice}$ = 23.7 μm, IWC = 5.0 ppmv and $RH_{ice}$ = 92%). (b): Corresponding $RH_{ice}$-$T_{am}$ relation. (c): $N_{ice}$–$R_{ice}$ relation for the cirrus outside aircraft plumes. (d): Corresponding $RH_{ice}$-$T_{am}$ relation. (e): $N_{ice}$–$R_{ice}$ relation for the cirrus fulfilling the plume, SAC and CA (ambient pressure 200–245hPa) criteria (median: $N_{ice}$ = 0.041 cm$^{-3}$, $R_{ice}$ = 17.8 μm, IWC = 4.4 ppmv and $RH_{ice}$ = 89%). (f): Corresponding $RH_{ice}$-$T_{am}$ relation.

**RC4**: Line 74, CONCERT 2018 campaign, should this be 2008?

**AC4**: This is a typo; it should be CONCERT 2008 campaign. It was corrected in the new version.

**References**

Giez, A., Mallaun, C., Zöger, M., Dörnbrack, A., and Schumann, U.: Static Pressure from Aircraft Trailing-Cone Measurements and Numerical Weather-Prediction Analysis, J. Aircr., 54, 1728-1737, https://doi.org/10.2514/1.C034084, 2017.

Kaufmann, S., Voigt, C., Heller, R., Jurkat-Witschas, T., Krämer, M., Rolf, C., Zöger, M., Giez, A., Buchholz, B., Ebert, V., Thornberry, T., and Schumann, U.: Intercomparison of midlatitude tropospheric and lower-stratospheric water vapor measurements and comparison to ECMWF humidity data, Atmos. Chem. Phys., 18, 16729-16745, https://doi.org/10.5194/acp-18-16729-2018, 2018.

Krämer, M., Rolf, C., Luebke, A., Afchine, A., Spelten, N., Costa, A., Meyer, J., Zöger, M., Smith, J., Herman, R. L., Buchholz, B., Ebert, V., Baumgardner, D., Borrmann, S., Klingebiel, M., and Avallone, L.: A microphysics guide to cirrus clouds – Part 1: Cirrus types, Atmos. Chem. Phys., 16, 3463-3483, https://doi.org/10.5194/acp-16-3463-2016, 2016.

Mallaun, C., Giez, A., and Baumann, R.: Calibration of 3-D wind measurements on a single-engine research aircraft, Atmos. Meas. Tech., 8, 3177-3196, https://doi.org/10.5194/amt-8-3177-2015, 2015.

Meyer, J., Rolf, C., Schiller, C., Rohs, S., Spelten, N., Afchine, A., Zöger, M., Sitnikov, N., Thornberry, T. D., Rollins, A. W., Bozóki, Z., Tátrai, D., Ebert, V., Kühnreich, B., Mackrodt, P., Möhler, O., Saathoff, H., Rosenlof, K. H., and Krämer, M.: Two decades of water vapor measurements with the FISH fluorescence hygrometer: a review, Atmos. Chem. Phys., 15, 8521-8538, https://doi.org/10.5194/acp-15-8521-2015, 2015.

Schumann, U.: Measurement and model data comparisons for the HALO-FAAM formation flight during EMeRGe on 17 July 2017, https://doi.org/10.5281/zenodo.4427965, 2021.

**Reply to Reviewer #2**

We thank Alexei Korolev for taking the time to carefully read through the manuscript and the insightful comments. Please find below the reviewer's comments in normal text, with our response in blue and the changes that has been made in the revised version of the manuscript in red.

**Comments**

**RC1**: Methodology: Identification of contrails embedded in cirrus and contrail cirrus clouds, within the P and T ranges, predetermined by CA, was based on the analysis of (a) the Schmidt-Appleman criterion (SAC) and (b) measurements of engine combustion products, aerosols and $NO_y$ (aircraft plume detection). A potential caveat of this approach is that $NO_y$ is a passive tracer, whereas cloud particles are an active cloud admixture in the atmosphere with a different response to the force of gravity and turbulent motions. As a result, at some point the contrail ice particles may become spatially separated from the plume and/or the plume may become spatially associated with particles formed in natural cirrus clouds. An explanation regarding this matter would clarify the limitations of the applied methodology. Specifically, what is the maximum age of contrail cirrus clouds when this method can be applied?

**AC1**: Thanks for the insightful comment, which we completely agree with, we have implemented these argumentations in the new version of the manuscript. The contrails that can be found with the help of the plume detection algorithm can be viewed as a subset of the contrails, because as long as the plume is detected and contains cirrus ice particles, these ice particles are highly likely to stem from contrails. We use the plume subset to show that the microphysical properties of these contrails are comparable to those from the larger data set (determined with SAC+CA) that also includes contrail ice particles spatially separated from the plume. This comparison increases our confidence in the method to identify contrails via the SAC+CA criterion; See Sect. 2.3.3 for the implemented changes in the revised version.

The maximum age of contrail cirrus clouds identified with the plume detection algorithm is estimated to be approximately 2–5 h, depending on the minimum $NO_y$ excess of around 0.1 ppbv above the atmospheric background, the diffusion speed of aircraft exhaust and $NO_y$ emission index of aircraft type.

**RC2**: As indicated in Table 1, the plume detection was only applied to approximately 2% of the collected data set. This brings up a question about the statistical significance of this data subset compared to data set with the SAC only criterion applied. It also would be relevant to state upfront in section 2.3.3 that the plume detection was applied only to a small fraction of the collected data, rather than having the reader figure it out after analysis of the data statistics in Table 1, at the end of the paper.

**AC2**: This is a good point, thank you. The plume detection algorithm is actually applied to the complete collected dataset. However, only a few percent of the air masses fit the criteria to be counted as an exhaust plume, for the reason discussed in RC1/AC1.

A table (see below) listing the size of dataset in flight time before and after applying different criteria (SAC, CA and plume detection) has been added to the revised Supplement of the paper as Table S2. And a statement of the plume detection algorithm including application restrictions ($NO_y$ concentration, approximate plume age rage, *etc.*) has been implemented to Sect. 2.3.3.

Table 1: In-cloud sampling time for different type of clouds after applying the Schmidt-Appleman criterion ($SAC^+$, fulfilling SAC), cruising altitude range ($CA^+$, in 200–245 hPa) and plume detection algorithm.

| Type of clouds | In-cloud time without plume detection | In-cloud time with plume detection |
|---|---|---|
| All type of clouds | 17.90 h | 1.04 h |
| All cirrus clouds ($T < 235$ K) | 14.70 h | 0.99 h |
| Contrail-natural cirrus mixture ($SAC^+$) | 11.18 h | 0.86 h |
| Contrail cirrus ($SAC^+$, $CA^+$) | 4.01 h | 0.35 h |

**RC3**: Airborne measurements of $RH_{ice}$ at temperatures below -50C are known to be of great challenge. It appears that the accuracy of the $RH_{ice}$ measurement required for the main outcomes of this paper should be of the order of 1%. Even though $RH_{ice}$ is one of the key parameters in this study, there are no discussions of the accuracy of measurements, inflight checks of the performance of humidity probes, etc. A brief discussion of this topic would be highly relevant in this paper, and it facilitate its reading rather than surfing through references. In this regard, I am wondering if you attempted inflight calibrations of water vapor probes in liquid clouds based on the methodology proposed in Korolev and Isaac (2006, JAS, https://doi.org/10.1175/JAS3784.1)?

**AC3**: Thanks again for pointing this out – the other referee (Minghui Diao) also criticised the lack of information on the accuracy of the $RH_{ice}$ measurement. The authors were probably too deeply involved to realize that the readers need more information. The overall uncertainty of SHARC $H_2O$ measurement is 5% relative and ±1 ppm absolute offset uncertainty (Kaufmann et al., 2018a). The nominal accuracies of the BAHAMAS pressure and $T_{amb}$ measurement are 0.3 hPa and 0.5 K (Mallaun et al., 2015; Giez et al., 2017; Kaufmann et al., 2018a). The overall accuracy of the in-situ $RH_{ice}$ measurements here is between $10-20\%$, with the respective uncertainties of the temperature, pressure and water vapour measurements considered (Krämer et al., 2016). This has been inserted into the instrument description in Sect. 2.1 Line 126.

Because SHARC $H_2O$ measurements agree well with other $H_2O$ instruments, such as AIMS which is operated with in-flight calibrations, in previous instrumental intercomparisons (Meyer et al., 2015; Kaufmann et al., 2018a), issues about the robustness of $RH_{ice}$ are not expected (This has been discussed in Sect 2.1 Line 130–133). However, in Sect 3.3, there is a brief discussion over the effect of a possible low temperature bias (which has only been raised recently by Schumann (2021)) on the $RH_{ice}$ distribution in relation to the ice-subsaturation feature in the contrail cirrus.

Unfortunately, inflight calibrations for SHARC are not possible, but the instrument is carefully calibrated before being deployed into the aircraft and after the complete of flight missions at the end of the campaign. For ML-CIRRUS, there were almost no measurements in liquid clouds, but looking into the $RH_w$ distribution in liquid clouds (T > 273 K) from a HALO campaign where more measurements are available (CIRRUS-HL) yields a sharp peak of $RH_w$ at 100% (see green curve in the figure below, which shows the occurrence frequency of $RH_i$ and $RH_w$ in mixed-phase clouds and $RH_w$ in liquid clouds). In mixed-phase clouds, as expected, $RH_w$ peaks in subsaturation (at 80%, turquoise), while the $RH_{ice}$ is also at 100% (dark cyan).

[Figure]

**RC4**: Section 4. I found the discussion around Figure 9 a bit misleading. The diagram in Figure 9 shows changes of T, $R_{ice}$, and $S_{ice}$ in an adiabatically ascending and then ascending parcel. The supersaturation in the vertically moving parcel will set to its quasi-steady value $S_{qs} = \frac{au_z}{N_{ice}\bar{r}_{ice}}$ at time $t > 3\tau_{ph}$, where $\tau_{ph}$ is the time of phase relaxation (see Korolev and Mazin, 2003, JAS, https://doi.org/10.1175/1520-0469(2003)060%3C2957:SOWVIC%3E2.0.CO;2). The two plateaus with $S_{qs}>0$ and $S_{qs}<0$ for the ascending and descending branches, respectively, are clearly visible in Fig.9. However, the authors consider only the descending branch, where the supersaturation is negative, and use it as an argument to explain the negative bias of $RH_{ice}$ in cirrus clouds. However, in stratiform type clouds, vertical ascending and descending motions are approximately equally probable, and the distribution $F(u_z)$ is typically centered

around 0. Keeping this in mind, and that $S_{qs}(u_z) = -S_{qs}(-u_z)$, the spatial averaging of humidity will yield $S \approx 0$.

In addition to the above, it is worth mentioning that complete evaporation of particles in adiabatic parcel will occur at the same level $Z_{ev}$, which depend on initial $IWC$ and the level $Z_0$. (To be strict, the level of complete sublimation depends on $u_z$. However, for the sake of argument, this effect of the condensational inertia can be neglected here.) Therefore, the lifetime of a descending cirrus parcel can be to a first approximation estimated as $t \sim (Z_0 - Z_{ev})/u_z$. Therefore, the estimated longevity of the subsaturated cirrus as 4h is a function of $u_z$ and $IWC(Z_0)$.

Having said the above, I would suggest reconsidering the argumentation in section 4 and the statement about 4h lifetime in the abstract.

**AC4**: The referee is right that Figure 9 shows an idealized ice cycle, including both the ascending and descending branches of an air parcel, *i.e.*, the cooling phase with $S_{qs}>0$ for ice formation and growth and the warming phase with $S_{qs}<0$ of ice crystals sedimenting into lower layers of the atmosphere. It's also true that in stratiform type clouds, vertical ascending and descending motions are approximately equally probable, and the distribution $F(uz)$ is typically cantered around 0.

As contrail cirrus formation process is not implemented in the model, the cooling phase of the simulated scenario is to produce cirrus particles that have the similar properties ($N_{ice}$ and $R_{ice}$) as the contrail cirrus identified in the paper. Driven by the vortex dynamics, the distribution of the vertical velocity in the wake of aircraft is distorted towards downdrafts, different from natural cirrus. The warming phase in Fig. 9 simulates the descending of contrails to several hundred meters below flight altitude, after the formation in primary aircraft vortex.

For contrail cirrus existing 90% $RH_{ice}$ environments to have an impact on the Earth's radiation, they should be at least persistent for a couple of hours. In an idealized case, the lifetime of the cirrus particles under constant cooling and IWC would be around 4 h. But in reality, as the referee said, the vertical velocity varies and ice crystals could exchange water vapour with surrounding ambient air, changing the IWC. The intention of this idealized ice cycle simulation with a focus on the warm phase is to let us have an assumption on the lifetime of the contrail cirrus in slightly ice subsaturated regions. In the revised version of the manuscript, we tried to describe better the simulation to stress the descent of contrails connected to

the wake vortex dynamics (starting from Line 561) and a reference to Sect. 3.3 where the contrail dynamics is discussed was added.

**RC5**: I attempted a simulation of the response of cirrus at $u_z = 0$ to the subsaturated environment with $RH_{ice}(0) = 90\%$, and the same $N_{ice}$ and $R_{ice}$ as indicated in Section 4. The results are shown in three diagrams to the right. It turned out that the in-cloud air arrives to saturation within ~25min. The red vertical line indicated $\tau_{ph}$ for initial $N_{ice}(0)$ and $R_{ice}(0)$. $\tau_{ph}$ shows a typical time of reaching saturation (usually within $3\tau_{ph}$). In this regard, it would be highly beneficial to indicate in Table 1 the time of phase relaxation.

[Figure]

[Figure]

**AC5**: Thanks to the referee for the suggestion. We made calculations of the phase relaxation time for the different cirrus groups listed in Table 1, using the equation (21) $S_{qsi} = \dfrac{1}{a_0 u_z + b_{i0} N_i \bar{r}_i}$ in Korolev and Mazin (2003). Under the conditions ($R_{ice}$, $N_{ice}$, $RH_{ice}$, temperature range shown in the table) where we observed the slightly subsaturated contrails, they would need ~30 min to relax to saturation when the descent is completed. The meaning and calculation of the phase relaxation time has been explained in Sect. 3.4, with the phase relaxation time of the different cirrus being indicated in Table 1.

[Figure]

**RC6**: IAGOS-MOSAIC data: I believe that the autonomous instruments installed in the commercial passenger aircraft in the frame of IAGOS were not maintained and calibrated with the same depth and frequency as on the HALO research airplane. Even though there are several references in the paper about the IAGOS data quality, it would be helpful to see a few general statements about the accuracy of $RH_{ice}$ measurements.

**AC6**: The deployment period of the IAGOS ICH sensors is usually 3 to 6 months. They are calibrated before and after the deployment. There is no online calibration for the measurements. The output signal (voltage) of sensors may drift during the period. Therefore, an in-flight calibration method is performed to all the reanalysis data by looking into the data from every 15 flights. During this process, the erroneous $RH_{ice}$ data caused by the drifts of sensor signals are corrected (Petzold et al., 2020; Smit et al., 2008). The limitation of detection of the ICH sensor is about 10% $RH_{ice}$, which might require more efforts in checking

the data in relatively dry lower stratosphere. The overall average uncertainty of the $RH_{ice}$ is about 5%, varying from 2% to 8% at 8–10 km cruising altitudes.

As the referee suggested, a general statement about the calibration, validation, uncertainty and LOD of IAGOS $RH_{ice}$ measurements were added into Sect. 2.2 Line 180–184.

**Minor comments**

**RC1**: Lines 13, 101, 266: It is not clear what the spatial statistics of the sampled clouds is. It is worth indicating the total length of sampled clouds along with the total cloud sampling time 14.7 h.

AC1: The authors are used to indicate cloud sample size using flight time. For more information for the readers, an estimation of the total length of sampled clouds converted from the aircraft maximum speed were indicated along the total cloud sampling time 14.7 h in Lines 13, 101 and the new line 264 in the revised version.

**RC2**: Line 141: In the equation for $R_{ice}$ the notations, "1.e4" and "1.e-6 " are confusing. It should be "$10^4$" and "$10^{-6}$".

**AC2:** "$10^4$" and "$10^{-6}$" in the new Line 147 in the revised version have replaced "1.e4" and "1.e-6 " in the equation in the original Line 141.

**RC3**: Section 2.1, Figure 6 and associated text: It would serve to clarify the paper to use the same type of definition of particle size, rather than switching between radius and diameter. Also indicate the definition of Dp., i.e., max particle size, average projected size, equivalent volume size, etc.

**AC3:** $D_p$ means optical equivalent diameter for CAS-DPOL and area equivalent diameter for CIP-Grayscale, respectively. This was added to Sect. 2.1 Line 139 and Fig. 6 caption. The measured particle size gives us more direct information of the particle size differences in addition to the mass mean radius.

**RC4**: Table 1. I found that IWC (mg/m3) calculated from $N_{ice}$ and $R_{ice}$ based on Eq. on line 141 is systematically lower than those indicated in Table 1. Was IWC (mg/m$^3$) calculated from IWC (ppmv)? A brief explanation in a footnote would be relevant.

**AC4:** IWC (mg/m$^3$) is converted from IWC (ppmv). This was noted in the revised manuscript in Line 147 and 527.

**RC5**: Figure 6b: The colors of PSDs for 'Contrail cirrus' and 'Contrail cirrus validated' appear to be the same (magenta and red). It is highly recommended to replace one of the colors by e.g., blue, violet, green, black for a better visualization of the curves.

**RC6**: Figure 7a: same as in #4.

**AC5-6:** Thank you for the recommendation. The magenta curves in Fig. 6b and 7a has been replaced by purple.

**RC7**: Figure 8: This diagram uses the same type of lines (i.e., dashed and solid) to indicate different curves.

**AC7**: Thank you for pointing it out. The line styles and legend in Fig. 8 have changed.

**RC8**: Line 651: "rather thin" => "rather optically thin".

**AC8**: "Rather thin" has been changed to "rather optically thin" in Line 622 in the revised version.

**References**

Giez, A., Mallaun, C., Zöger, M., Dörnbrack, A., and Schumann, U.: Static Pressure from Aircraft Trailing-Cone Measurements and Numerical Weather-Prediction Analysis, J. Aircr., 54, 1728-1737, https://doi.org/10.2514/1.C034084, 2017.

Kaufmann, S., Voigt, C., Heller, R., Jurkat-Witschas, T., Kramer, M., Rolf, C., Zoger, M., Giez, A., Buchholz, B., Ebert, V., Thornberry, T., and Schumann, U.: Intercomparison of midlatitude tropospheric and lower-stratospheric water vapor measurements and comparison to ECMWF humidity data, Atmos. Chem. Phys., 18, 16729-16745, https://doi.org/10.5194/acp-18-16729-2018, 2018a.

Kaufmann, S., Voigt, C., Heller, R., Jurkat-Witschas, T., Krämer, M., Rolf, C., Zöger, M., Giez, A., Buchholz, B., Ebert, V., Thornberry, T., and Schumann, U.: Intercomparison of midlatitude tropospheric and lower-stratospheric water vapor measurements and comparison to ECMWF humidity data, Atmos. Chem. Phys., 18, 16729-16745, https://doi.org/10.5194/acp-18-16729-2018, 2018b.

Korolev, A. V. and Mazin, I. P.: Supersaturation of water vapor in clouds, J. Atmos. Sci., 60, 2957-2974, https://doi.org/10.1175/1520-0469(2003)060<2957:Sowvic>2.0.Co;2, 2003.

Krämer, M., Rolf, C., Luebke, A., Afchine, A., Spelten, N., Costa, A., Meyer, J., Zöger, M., Smith, J., Herman, R. L., Buchholz, B., Ebert, V., Baumgardner, D., Borrmann, S., Klingebiel, M., and Avallone, L.: A microphysics guide to cirrus clouds – Part 1: Cirrus types, Atmos. Chem. Phys., 16, 3463-3483, https://doi.org/10.5194/acp-16-3463-2016, 2016.

Mallaun, C., Giez, A., and Baumann, R.: Calibration of 3-D wind measurements on a single-engine research aircraft, Atmos. Meas. Tech., 8, 3177-3196, https://doi.org/10.5194/amt-8-3177-2015, 2015.

Meyer, J., Rolf, C., Schiller, C., Rohs, S., Spelten, N., Afchine, A., Zöger, M., Sitnikov, N., Thornberry, T. D., Rollins, A. W., Bozóki, Z., Tátrai, D., Ebert, V., Kühnreich, B., Mackrodt, P., Möhler, O., Saathoff, H., Rosenlof, K. H., and Krämer, M.: Two decades of water vapor measurements with the FISH fluorescence

hygrometer: a review, Atmos. Chem. Phys., 15, 8521-8538, https://doi.org/10.5194/acp-15-8521-2015, 2015.

Petzold, A., Neis, P., Rütimann, M., Rohs, S., Berkes, F., Smit, H. G. J., Krämer, M., Spelten, N., Spichtinger, P., Nédélec, P., and Wahner, A.: Ice-supersaturated air masses in the northern mid-latitudes from regular in situ observations by passenger aircraft: vertical distribution, seasonality and tropospheric fingerprint, Atmos. Chem. Phys., 20, 8157-8179, https://doi.org/10.5194/acp-20-8157-2020, 2020.

Schumann, U.: Measurement and model data comparisons for the HALO-FAAM formation flight during EMeRGe on 17 July 2017, https://doi.org/10.5281/zenodo.4427965, 2021.

Smit, H. G. J., Volz-Thomas, A., Helten, M., Paetz, W., and Kley, D.: An in-flight calibration method for near-real-time humidity measurements with the airborne MOZAIC sensor, J. Atmos. Ocean. Technol., 25, 656-666, https://doi.org/10.1175/2007JTECHA975.1, 2008.